# Microbial recognition by GEF-H1 controls IKKε mediated activation of IRF5

Yun Zhao[1], Rachid Zagani[1], Sung-Moo Park [1], Naohiro Yoshida[1], Pankaj Shah[1] & Hans-Christian Reinecker[1]

During infection, transcription factor interferon regulatory factor 5 (IRF5) is essential for the control of host defense. Here we show that the microtubule-associated guanine nucleotide exchange factor (GEF)-H1, is required for the phosphorylation of IRF5 by microbial muramyl-dipeptides (MDP), the minimal structural motif of peptidoglycan of both Gram-positive and Gram-negative bacteria. Specifically, GEF-H1 functions in a microtubule based recognition system for microbial peptidoglycans that mediates the activation of IKKε which we identify as a new upstream IKKα/β and IRF5 kinase. The deletion of GEF-H1 or dominant-negative variants of GEF-H1 prevent activation of IKKε and phosphorylation of IRF5. The GEF-H1-IKKε-IRF5 signaling axis functions independent of NOD-like receptors and is critically required for the recognition of intracellular peptidoglycans and host defenses against *Listeria monocytogenes*.

[1] Department of Medicine, Gastrointestinal Unit and Center for the Study of Inflammatory Bowel Disease, Massachusetts General Hospital, Harvard Medical School, Boston, MA 02114, USA. These authors contributed equally: Yun Zhao, Rachid Zagani. Correspondence and requests for materials should be addressed to H.-C.R. (email: hans-christian_reinecker@hms.harvard.edu)

Microtubules are components of the cytoskeleton that have important functions in cell-autonomous and innate immunity to enable antimicrobial host defense in addition to their roles in control of cell division shape and movement[1]. The guanine nucleotide exchange factor-H1 (GEF-H1), encoded by *ARHGEF2*, is a microtubule associated protein that is crucial for sensing intracellular pathogens and initiating transcriptional programs that counter bacterial and viral infections[2,3].

IRF5 is a critical transcription factor regulating immune and inflammatory responses in host defense and disease. Further, polymorphisms in the IRF5 gene have been linked to human autoimmune diseases[4–6]. In addition, IRF5 plays important roles in inflammatory M1-like macrophage polarization[7]. IRF5 has an essential role in the control of inflammation initiated by the activation of RIPK2 in NOD-like receptor (NLR) and IKKβ during Toll-like (TLR) signaling[8,9]. IKKβ together with IKKα also mediates NF-κB activation by phosphorylating IκB proteins as part of the IKK complex together with the regulatory subunit termed IKKγ[10–13].

IKK-related kinases, IKKε[14,15] and TBK1[16–18], have been linked to the activation of IRF3 and IRF7 which function as important mediators of antiviral host responses. We have previously shown that GEF-H1 can interact with TBK1 for the activation IRF3 during MAVS-dependent RLR receptor signaling[3]. However, host defense circuits that specifically activate either IKKε or TBK1 have not been specified.

This work uncovers a microtubule based recognition system for microbial peptidoglycan where GEF-H1 is essential for activating the IKK-related kinases, IKKε which we identify as a previously unrecognized IKKα/β and IRF5 kinase. Upon release from microtubules the GEF-H1 signalosome mediates the activation of Rho-Associated, Coiled-Coil Containing Protein Kinases (ROCK) and forms protein complexes that assembled IRF5, IKKε, and IKKα/β. Although the ability of GEF-H1 to specify IKKε phosphorylation is specific to MDP recognition, IKKε also mediates IRF5 phosphorylation during Toll-like receptor signaling. GEF-H1 mediated IRF5 activation by MDP occurs independent of NOD2. The newly proposed GEF-H1-IKKε-IRF5 signaling axis controls transcriptional programs activated by MDP that define peptidoglycan receptor expression and genes that control intracellular host defense against *L. monocytogenes* in macrophages.

## Results

### GEF-H1 is essential for phosphorylation of IRF5 by MDP.
To determine the role of GEF-H1 in signaling events induced by MDP, we isolated bone marrow-derived macrophages (BMDMs) from wild-type (WT) C57BL/6, *Arhgef2*$^{-/-}$, and *Nod2*$^{-/-}$ mice, and determined the phosphorylation and nuclear translocation of IRF5 after stimulation with MDP. As demonstrated in Fig. 1a, lack of GEF-H1 expression impaired IRF5 phosphorylation during MDP-induced immune activation of WT macrophages (Fig. 1a). Surprisingly, IRF5 phosphorylation in response to MDP stimulation occurred unimpeded in *Nod2*$^{-/-}$ mice (Fig. 1a). MDP induced *Nod2* mRNA expression in WT and *Arhgef2*$^{-/-}$ but not in *Nod2*$^{-/-}$ deficient macrophages within 4 h (Fig. 1b). *Arhgef2*$^{-/-}$ macrophages also demonstrated less nuclear translocation of IRF5 compared to WT and *Nod2*$^{-/-}$ macrophages upon MDP stimulation (Fig. 1c). GEF-H1 and IRF5 may be able to directly interact as anti-IRF5 but not control IgG pulled down immunocomplexes that contained GEF-H1 from WT but not *Arhgef2*$^{-/-}$ macrophages in the absence or presence of MDP (Fig. 1d). IRF5 also became part of protein complexes that were pulled down with anti-GEF-H1 in the presence or absence of

RIPK2 when expressed in HEK293 cells (Fig. 1e). GEF-H1 can interact with RIPK2 which has been identified as a kinase upstream of IKKβ for the activation of NF-κB and IRF5[19] and we indeed found that *Ripk2*$^{-/-}$ macrophages failed to induce IRF5 phosphorylation in response to MDP (Fig. 1f). Furthermore, RIPK2 function was required in MDP stimulated macrophages for the phosphorylation of IKKε and IKKα/β at S176/180 or S172, respectively (Fig. 1g). We further found that GEF-H1 was able to directly interact with IKKε and IKKβ in the absence or presence of NOD2 in HEK293T cells (Fig. 1h). Together these data demonstrated that GEF-H1 interacted with RIPK2, IKKε, and IKKβ for the recruitment and phosphorylation of IRF5 during MDP recognition.

### IKKε is required for the phosphorylation of IRF5 by IKKα/β.
Since GEF-H1 interacted with both IKKε and IKKβ, we sought to determine whether IKKε had a role in the phosphorylation of IRF5 during MDP induced cell signaling. We observed enhanced phosphorylation of IRF5 when IKKε but not a kinase deficient IKKε (K38A) variant was co-expressed with GEF-H1 (Fig. 2a). This indicated that GEF-H1 enabled IKKε to function as an upstream IFR5 kinase. As demonstrated in Fig. 2b, IKKε was indeed an essential upstream kinase for the phosphorylation of IRF5 but not NF-κB p65 in response to MDP in macrophages. *Ikkε*$^{-/-}$ macrophages were unable to respond to MDP stimulation with the phosphorylation of S445 of IRF5 while phosphorylation of p65 was detectable (Fig. 2b). Further, *Ikkε*$^{-/-}$ macrophages failed to activate IKKα/β by phosphorylating S176/180 in response to MDP (Fig. 2c). In the MDP induced signaling pathway, however, neither TBK1 nor IRF3 were significantly phosphorylated when compared to cyclic di GMP-induced STING signaling (Supplementary Fig. 1). Together these results indicated that IKKε could be a target of RIPK2 for IKKα/β phosphorylation during GEF-H1-dependent IRF5 phosphorylation. Indeed, GEF-H1 and RIPK2 together significantly enhanced IKKε phosphorylation compared to RIPK2 and GEF-H1 expression alone when co-expressed in HEK-293T cells (Fig. 2d). We next determined whether IKKε functioned downstream of RIPK2 in the phosphorylation of IRF5 by overexpressing RIPK2 in the *Ikkε*$^{-/-}$ (Fig. 2e) and IKKε in *Ripk2*$^{-/-}$ macrophages (Fig. 2f). RIPK2 induced significant less phosphorylation of IRF5 in *Ikkε*$^{-/-}$ compared to WT macrophages (Fig. 2e) while overexpression of IKKε in *Ripk2*$^{-/-}$ macrophages elicited levels of IRF5 phosphorylation that were comparable to those induced in WT macrophages (Fig. 2f). As GEF-H1 is not involved in mediating TLR4 signal transduction[3], IRF5 phosphorylation occurred comparable in WT and *Arhgef2*$^{-/-}$ macrophages in response to LPS (Fig. 2g). In response to LPS, we also observed the activation of NF-κB in *Arhgef2*$^{-/-}$ or *Ikkε*$^{-/-}$ macrophages (Fig. 2g). However, IKKε was also required for IRF5 phosphorylation at S445 in response to LPS stimulation of macrophages demonstrating a convergence of distinct pattern recognition pathways activating IKKε for IRF5 phosphorylation (Fig. 2g). Together, these experiments indicated that GEF-H1 specified IKKε function for IKKα/β-mediated IRF5 phosphorylation by MDP.

### GEF-H1 dephosphorylation is required for IKKε and IRF5 binding.
In the next set of experiments, we aimed to define the dephosphorylation events and amino acids of GEF-H1 that are required for the interaction with IKKε and IRF5. Sequence analysis of full-length human GEF-H1 (positional information refers to NP_001155855.1) revealed a pLxIS (p, hydrophilic residues; x, any amino acid) consensus motif in the C-terminal regions of GEF-H1 at amino acids 320–324 (Fig. 3a). This motif has been proposed to mediate interaction among IRF3, MAVS, and

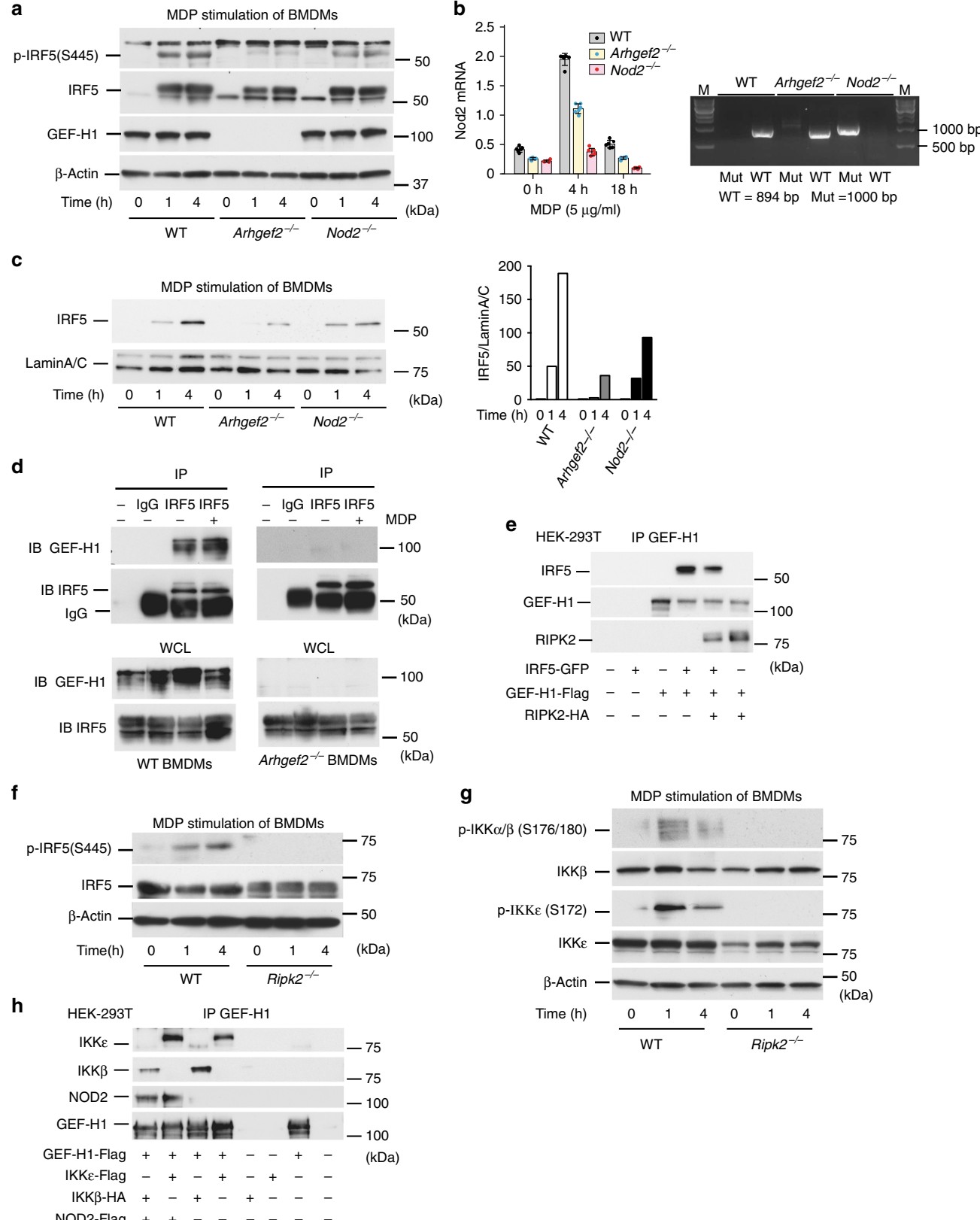

STING[20]. The pLxIS motif in IRF5 has been shown to contain a Serine that is phosphorylated by IKKβ in response to LPS and is detected by our antibody[9,21]. GEF-H1 contains a second related motif YPLxIS at amino acids 394–399 in which the charged residue in the pLxIS motif is replaced by a Proline but preceded by the Tyrosine at position 394 that is required for exchange function and IRF3 phosphorylation by GEF-H1 in the RIG-I-like receptor signaling[3]. The addition of a phosphate molecule to a non-polar R group of an amino acid residue could turn a hydrophobic portion of a protein into a polar and extremely hydrophilic one. Figure 3a summarizes the variants created by exchanging the Serines or Tyrosines, respectively, at amino

**Fig. 1** GEF-H1 mediates IRF5 phosphorylation during MDP recognition. **a** Immunoblot analysis of the activation of IRF5 with antibodies detecting phosphorylated (p-) or total IRF5 in whole-cell lysate of bone marrow derived macrophages (BMDMs) from WT, *Arhgef2*−/− and *Nod2*−/− mice after stimulation with *N*-glycolyl-MDP (5 μg ml−1) for 1 h and 4 h. β-actin used as loading control. **b** *Nod2* mRNA expression analysis by qRT-PCR from WT, *Arhgef2*−/− and *Nod2*−/− derived BMDMs and genotyping PCR for targeted or wildtype *Nod2* allele in genomic DNA from *Nod2*−/−, WT and *Arhgef2*−/− mice. **c** Immunoblot analysis of IRF5 expression in nuclear extracts from BMDMs from WT, *Arhgef2*−/− and *Nod2*−/− mice after stimulation with *N*-glycolyl-MDP with band intensities quantified using densitometry and Image J software. **d** Immunoblot analysis of GEF-H1 expression after IRF5 pull down in BMDMs from WT and *Arhgef2*−/− mice in presence or absence of *N*-glycolyl-MDP. **e** Immunoprecipitation of GEF-H1 from HEK293T cells that were transfected with IRF5, GEF-H1 and/or RIPK2 encoding plasmids. **f** Western blot analysis of the expression and phosphorylation of IRF5 in BMDMs from WT and *Ripk2*−/− mice that were stimulated with *N*-glycolyl-MDP for 1 h and 4 h. **g** Immunoblot analysis of the expression and phosphorylation of IKKα/β and IKKε in BMDMs from WT and *Ripk2*−/− mice after stimulation with *N*-glycolyl-MDP for 1 h and 4 h. **h** Immunoprecipitation of GEF-H1 with specific ABs from HEK293T cells that were transfected with GEF-H1-Flag, IKKε-Flag, **IKKβ**-HA, and NOD2-Flag expression constructs. Anti-GEF-H1, anti-IKKε, and anti-IKKβ were used to detect proteins by western blotting and NOD2 was detected with anti-Flag. Source data are provided as a Source Data file

acid 324, 394, 399, and S886 of GEF-H1 to define their relevance for binding and phosphorylating IKKε, IKKβ, and IRF5.

Initially, we determined the phosphorylation status of GEF-H1 variants in HEK293T cells in the absence or presence of IKKε and IKKβ. Surprisingly, modifying Serine 399 of GEF-H1 resulted in a significantly elevated phosphorylation of GEF-H1 at S886 as detected by a specific antibody for this phosphorylated residue (Fig. 3b). Also modifying S324 elevated baseline S886 phosphorylation of GEF-H1, although significantly less compared to S399 (Fig. 3b). The phosphorylation of GEF-H1 at S886 may have important functional consequences, as de-phosphorylation of this residue is required for the release of GEF-H1 from microtubules[22]. IKKε but not IKKβ further induced additional phosphorylation events that occurred on GEF-H1 and GEF-H1 (Y394A) as indicated by a pan phosphor-serine/threonine antibodies (Fig. 3c).

Remarkably, the hyper-phosphorylated GEF-H1 (S399A) variant was unable to bind IKKε while the exchange deficient GEF-H1 mutant Y394A was able to bind IKKε efficiently (Fig. 3d). IKKε binding was also reduced to the GEF-H1 (S324A) variant further indicating that activation of GEF-H1 by dephosphorylation was required to facilitate efficient IKKε binding (Fig. 3d). In contrast, none of these GEF-H1 variants prevented IKKβ binding to GEF-H1 (Supplementary Fig. 2). To prevent conformational changes in GEF-H1 upon S886 phosphorylation that could interfere with IKKε binding, we created GEF-H1 variants with double substitution of Alanine for S886 and S324 or S399. Co-immunoprecipitations demonstrated that removing the phosphorylation event at S886 resulted in the increased binding of IKKε to GEF-H1 (Fig. 3e). While the GEF-H1 variant S399A was unable to bind IKKε, the S399A/S886A double mutant interacted with IKKε indicating that binding to GEF-H1 was required for the activation of IKKε by phosphorylation at S172 (Fig. 3e). However, IKKε binding to the GEF-H1 S324A variant also increased when the phosphorylation event at S886 was removed from GEF-H1. GEF-H1 variants that were able to bind IKKε indeed induced the phosphorylation of S172 of IKKε. (Fig. 3e). These experiments also revealed that S324 was particularly important in controlling IKKε phosphorylation as its removal resulted in enhanced phosphorylation of IKKε bound to GEF-H1 (Fig. 3e).

We next determined whether modifying S324 or S399 alone, or in combination with S886 controlled IRF5 binding of GEF-H1. Preventing phosphorylation at S886 resulted in enhanced IRF5 binding to GEF-H1. IRF5 binding was further enhanced to a GEF-H1 variant with Alanine substitutions for S886 and S324. In contrast, exchanging Alanine for S399 in the S886A variant failed to enhance IRF5 binding to GEF-H1 (Fig. 3f). Together, these data demonstrated that the innate immune function of GEF-H1 is controlled by dephosphorylation events that allow the interaction with and phosphorylation of IKKε. Both, S324 and S886 of GEF-H1 are critical in the control of the interaction with IRF5. Further,

S399 within the YPLxIS domain of GEF-H1 plays an important role for the control of phosphorylation events at S886 that inactivate GEF-H1 immune function.

**The GEF-H1-IKKe-IRF5 pathway is controlled by ROCK1/2.** We co-expressed RFP tagged GEF-H1 and GFP tagged IRF5 together with either IKKε or the kinase deficient IKKε K38A variant to determine subcellular localization of GEF-H1 and nuclear translocation of IRF5 during cell autonomous immune activation by confocal microscopy. GEF-H1 localized to the microtubule network and bundles in the presence of the IKKε K38A variant with IRF5 that was distributed throughout the cytoplasm and enriched in large subcellular compartments (Fig. 4a). When we activated the pathway by expressing functional IKKε, GEF-H1 lost association with the microtubule network and co-localized with IRF5 in small subcellular compartments throughout the cytoplasm (Fig. 4a). Upon co-expression of GEF-H1 together with IKKε, IRF5 translocated to the nucleus (Fig. 4a). Cells that demonstrated nuclear translocation of IRF5 also exhibited membrane blebbing indicative of Rho GTPase function that is enhanced by GEF-H1 and leads to Rho-associated protein kinase (ROCK1) activation[2,23].

We therefore determined whether GEF-H1 was able to control the phosphorylation of ROCK1 in the context of IKKε. In these experiments we immunoprecipitated ROCK1 from HEK293 cells that were transfected with ROCK1, IKKε, and wild type, and variants of GEF-H1 to determine ROCK1 phosphorylation at S455/T456. Remarkably, the GEF-H1 double-mutant S324A/S886A allowed the highest phosphorylation of ROCK1 while the S399A/S886A variant failed to induce significant ROCK1 phosphorylation (Fig. 4b). The GEF-H1 variants S324A, S399A, and S886A were able to confer some ROCK1 phosphorylation in the presence of IKKε when compared to wild-type GEF-H1 or the exchange deficient mutant GEF-H1Y394H (Fig. 4b). These experiments indicated that dephosphorylation of GEF-H1 may allow the activation of ROCK1 as a requirement for IRF5 phosphorylation. Indeed, ROCK1 mediated phosphorylation of IKKε and IRF5 in the presence of functional IKKε but not inactive IKKε K38A (Fig. 4c). ROCK1 likely functioned downstream of GEF-H1 as during ROCK1 overexpression, GEF-H1 remained associated with microtubules while nuclear IRF5 localization occurred (Fig. 4d). Finally, the ROCK1/2 inhibitor Y27632 inhibited MDP-induced immune activation as IRF5 as well as NF-κB phosphorylation was reduced in the presence of the inhibitor in WT macrophages (Fig. 4e). Collectively, these data indicated that IKKε and IRF5 activation by GEF-H1 was controlled through ROCK1/2.

**GEF-H1 and IRF5 control MDP-initiated transcription.** To identify the MDP-induced transcriptional responses that

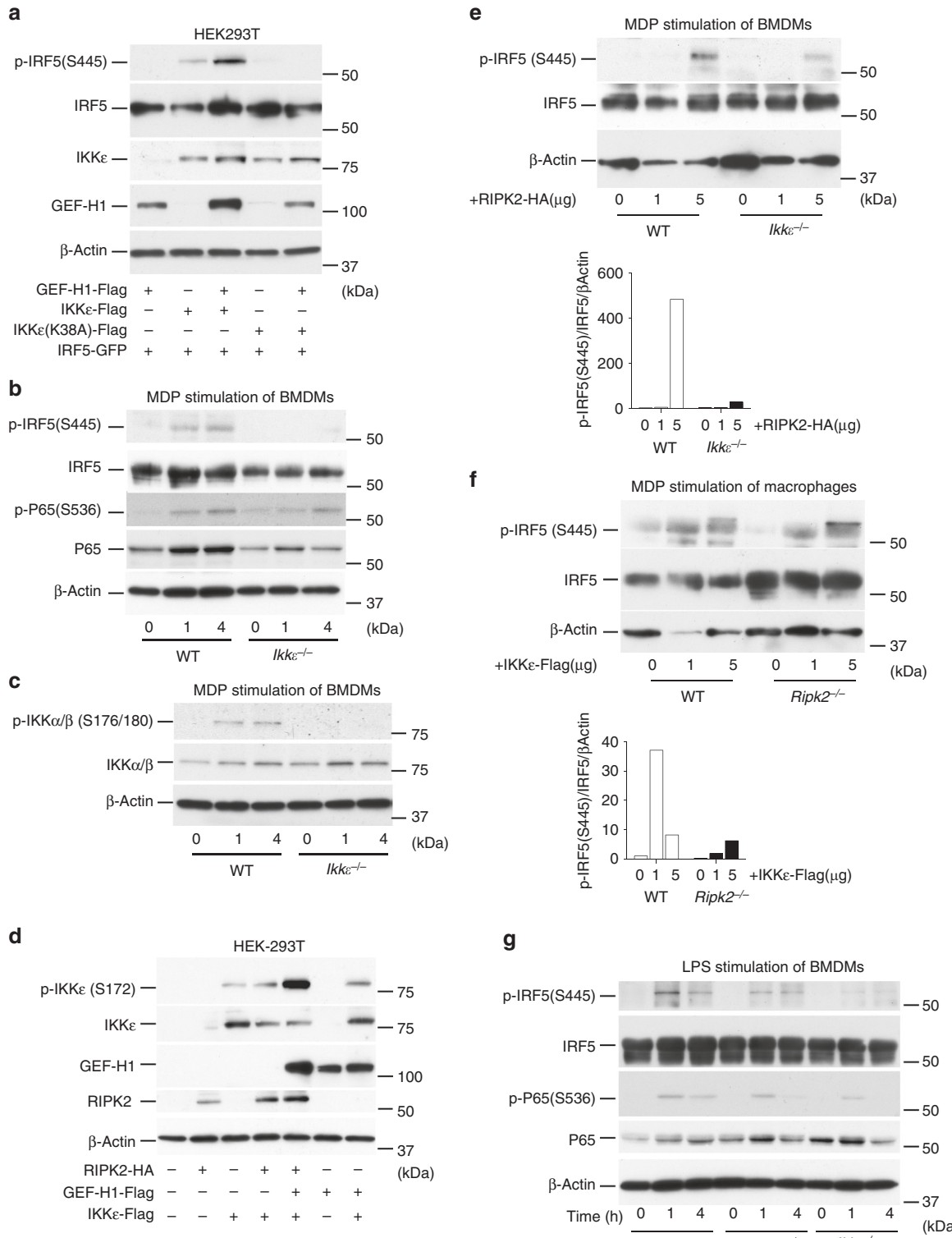

**Fig. 2** IKKε is required for the phosphorylation of IRF5. **a** Assessment of IRF5 phosphorylation in the absence or presence of GEF-H1, IKKε, or a kinase deficient IKKε (K38A) variant in HEK293T cells. **b** Immunoblot analysis of IRF5 and NF-κB-p65 phosphorylation in BMDMs from WT and *Ikkε*⁻/⁻ mice after stimulation with *N*-glycolyl-MDP for 1 h and 4 h. **c** Western blot analysis of IKKα/β phosphorylation in BMDMs from WT and *Ikkε*⁻/⁻ mice after stimulation with *N*-glycolyl-MDP for 1 h and 4 h. **d** Immunoblot analysis of IKKε phosphorylation mediated by GEF-H1 and/or RIPK2 after co-expression in HEK293T cells. **e** Immunoblot analysis of IRF5 phosphorylation in response to RIPK2 overexpression in WT and *Ikkε*⁻/⁻ BMDMs with band intensities quantified using densitometry and Image J software. **f** Western blot analysis of IRF5 phosphorylation in response to overexpression of IKKε in WT or *Ripk2*⁻/⁻ macrophages with band intensities quantified using densitometry and Image J software. **g** Assessment of expression and phosphorylation of IRF5 and NF-κB-p65 in BMDMs from WT, *Arhgef2*⁻/⁻, and *Ikkε*⁻/⁻ mice after exposure to LPS (100 ng ml⁻¹) for 1 h and 4 h. Source data are provided as a Source Data file

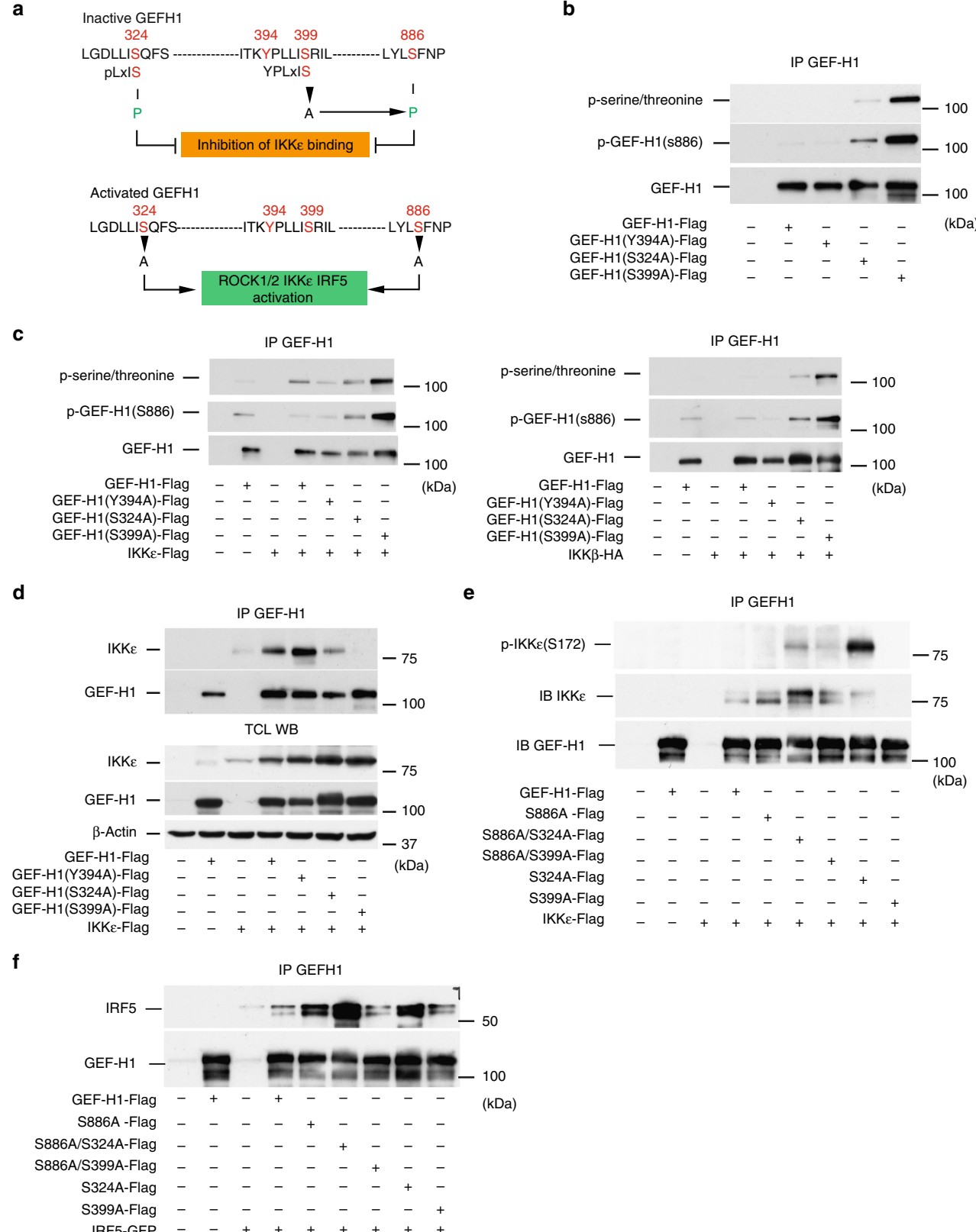

specifically require GEF-H1 and IRF5, we performed high-resolution mRNA expression profiling using next-generation RNA sequencing (RNA-seq). We used samples of RNA isolated from BMDMs of *Arhgef2*<sup>−/−</sup>, *Irf5*<sup>−/−</sup>, and WT mice that were stimulated for 18 h with MDP. RNA-seq pipeline quantitation in Seqmonk was used on merged transcripts counting reads over exons and log transformed. Principal component analysis (PCA) of normalized expression revealed that control and MDP-treated WT BMDMs segregated into distinct quartiles, and the control and treated BMDMs lacking *Arhgef2* or *Irf5* remained clustered close together indicating a reduced transcriptional response (Fig. 5a). The lack of transcriptional changes in response to MDP

**Fig. 3** GEF-H1 dephosphorylation activates IKKε and IRF5 binding. **a** Representation of GEF-H1 amino acid motifs and targeted residues that control dephosphorylation of GEF-H1 for binding and activating IKKε and IRF5. **b** WT or GEF-H1 variants were expressed in HEK293T cells and the phosphorylation of GEF-H1 assessed after immunoprecipitation with anti-GEF-H1 with pan-phospho S/T ABs or ABs detecting the phosphorylation of GEF-H1 at S886. **c** Phosphorylation of GEF-H1 and indicated GEF-H1 variants was assessed in HEK293T after immunoprecipitation with anti-GEF-H1 in the presence of either IKKε or IKKβ with anti-pan-phospho S/T antibodies or ABs detecting the phosphorylation of GEF-H1 at S886. **d** WT or GEF-H1 variants and IKKε were co-expressed in HEK293T cells, GEF-H1 was pulled down with anti-GEF-H1 and co-immunoprecipitated IKKε detected by western blotting. **e** Amount of phosphorylated and total IKKε that was bound to immunoprecipitated WT and GEF-H1 variants after expression in HEK293T cells. **f** Co-immunoprecipitation of IRF5 with anti-GEF-H1 antibody in the presence of WT GEF-H1 or GEF-H1 variants expressed in HEK293T cells. Source data are provided as a Source Data file

in *Arhgef2*$^{-/-}$ and *Irf5*$^{-/-}$ BMDMs was also revealed in pairwise comparisons in a Cuffdiff analysis where the majority of mRNAs that were upregulated more than twofold by MDP in WT BMDMs failed to transcriptionally activate in *Arhgef2*$^{-/-}$ and *Irf5*$^{-/-}$ BMDMs (Fig. 5b). Hierarchical clustering of MDP regulated genes revealed that GEF-H1 and IRF5 were required for a significant proportion of the transcriptional response induced by MDP (Fig. 5b). Out of the 984 regulated genes with more than twofold regulation upon MDP stimulation in WT BMDMs, GEF-H1, and IRF5 were both required for the induction of 422 or inhibition of 36 transcripts (Fig. 5b, Cluster I and VII, Supplementary Data 1). We observed additional gene clusters that were either dependent on GEF-H1 (cluster IV, 43 genes, Supplementary Data 1) or IRF5 alone (cluster II, 77 genes, Supplementary Data 1). We also detected an additional cluster containing 210 transcripts that were induced by MDP independently of GEF-H1 and IRF5 (Fig. 5b Clusters III and; Supplementary Data 1). For Gene Set Enrichment Analyses (GSEA) genes were further ranked by intensity difference based on their dependence on GEF-H1 or IRF5 and the extent of regulation upon MDP stimulation. Figure 5c shows the transcripts in cluster I with a Z-score >3.0 that required both GEF-H1 and IRF5 for induction by MDP. GSEA analysis associated the GEF-H1 and IRF5 dependent gene clustered in the MSigDB Hallmark data base significantly with mesenchymal transition, TNF-α signaling, and Inflammatory responses, and in the MSigDB Canonical pathway (KEGG) database with Focal adhesion, ECM, and cytokine - cytokine receptor interaction (Fig. 5d).

In the next set of experiments, we analyzed the expression of antimicrobial host defense genes induced by MDP in Cluster 1 in macrophages isolated from WT, *Arhgef2*$^{-/-}$ *Irf5*$^{-/-}$ and *Ikkε*$^{-/-}$ mice by qRT-PCR. These experiments confirmed that GEF-H1, IKKε, and IRF5 mediated the expression of peptidoglycan recognition protein 1 (*Pglyrp1*; Fig. 5e). Pglyrps recognize bacterial cell wall peptidoglycans of Gram-positive and Gram-negative bacteria and have antibacterial functions[24–26]. Additional antimicrobial factor co-regulated by GEF-H1, IKKε, and IRF5 included GranzymeD (*GzmD*), GranzymeE (*GzmE*), and *Serpine1* all of which have been shown to mediate host defenses[27,28] (Fig. 5e). Together, these experiments indicated that GEF-H1 and IKKε mediated phosphorylation of IRF5 controlled a unique transcriptional program that could initiate innate immune responses critical for anti-microbial host defense in macrophages through the detection of peptidoglycans and microbial elimination.

**The GEF-H1-IKKε-IRF5 signaling axis controls *L. mono-cytogenes*.** We next determined the role of GEF-H1, IKKε, and IRF5 in the activation of host defense against *L. monocytogenes*, the causative agent of listeriosis that controls the host-cell cytoskeleton for invasion and intracellular spread[29]. The recognition of *L. monocytogenes* requires peptidoglycan[30] and nucleic acid sensing receptors[31]. During infection of ARPE-19 cells with GFP expressing *L. monocytogenes*, RFP-GEFH1 fusion protein

redistributed from microtubules into the cytoplasm (Fig. 6a). Exposure to *L. monocytogenes* induced IRF5 phosphorylation in WT mice but not in *Arhgef2*$^{-/-}$ or *Ikkε*$^{-/-}$ macrophages (Fig. 6b). *Ikkε*$^{-/-}$ macrophages were also unable to respond to *L. monocytogenes* with the same level of phosphorylation of NF-κB-p65 that occurred in WT and *Arhgef2*$^{-/-}$ macrophages (Fig. 6b). The impaired innate immune activation in macrophages derived from *Arhgef2*$^{-/-}$, *Ikkε*$^{-/-}$, or *Irf5*$^{-/-}$ mice had a significant impact on antimicrobial defenses. Macrophages that lacked GEF-H1, IKKε, or IRF5 were significantly more susceptible to infection with *L. monocytogenes* (Fig. 6c). Exposure to $10^5$ bacteria over 2 h resulted in a 6.9-fold higher bacterial load in *Arhgef2*$^{-/-}$, 4.3-fold in *Ikkε*$^{-/-}$, and 7.9-fold in *Irf5*$^{-/-}$ macrophages (Fig. 6c). Furthermore, bacterial elimination was particularly lacking in GEF-H1-deficient macrophages which were unable to significantly reduce bacterial burden 6 h after uptake in contrast to WT macrophages. *Ikkε* and *Irf5* deficient macrophages were able to reduce intracellular *L. monocytogenes* by 53 and 74%, respectively. We next determined whether *L. monocytogenes* induced the transcriptional regulation of genes that were IRF5 and GEF-H1 dependently induced by MDP in macrophages. Indeed, *L. monocytogenes* induced the expression of mRNA encoding for *Pglyrp1*, *GzmD*, and *GzmE*, and *Serpine1* in WT macrophages. The induction of these genes during *L. monocytogenes* infection was significantly reduced in macrophages that were *Arhgef2*, *Ikkε*, or *Irf5* deficient (Fig. 6d). Together, these data showed that the GEF-H1-IKKε-IRF5 signaling axis was activated as a detection system that defines antimicrobial host defenses in macrophages.

## Discussion

We here demonstrate that GEF-H1 is a critical activator and signaling platform for microbial peptidoglycans recognition that is essential for the phosphorylation of IRF5 by the atypical IKK kinase IKKε. We propose a model in which a GEF-H1 signalosome recruits ROCK1, IKKε, and IRF5, as a prerequisite for IRF5 phosphorylation in response to intracellular peptidoglycan recognition. IKKε is the second atypical IKK kinase that can bind GEF-H1. In the MAVS pathway, GEF-H1 can interact with TBK1 and mediate the phosphorylation of IRF3 and induction of type 1 interferons[3]. However, during the recognition of MDP, GEF-H1 interacted specifically with IKKε for the phosphorylation of IRF5 demonstrating that despite substantial sequence homology the functions of IKKε and TBK1 can be defined by distinct substrate specificities and signaling intermediaries that are immune recognition pathway specific. Previously, it had been difficult to distinguish specific functions of IKKε and TBK1 as they are often simultaneously activated during pathogen-associated molecular pattern recognition[32].

Our mutational analysis indicates that phosphorylation events at S324, S399, and S886 are critical for the function of GEF-H1. Remarkably, preventing phosphorylation at S399 resulted in the hyper-phosphorylation of GEF-H1 at S886 which prevented the binding to IKKε required for IKKε phosphorylation. Thus, S399 maybe critical for mediating interaction with a phosphatase that

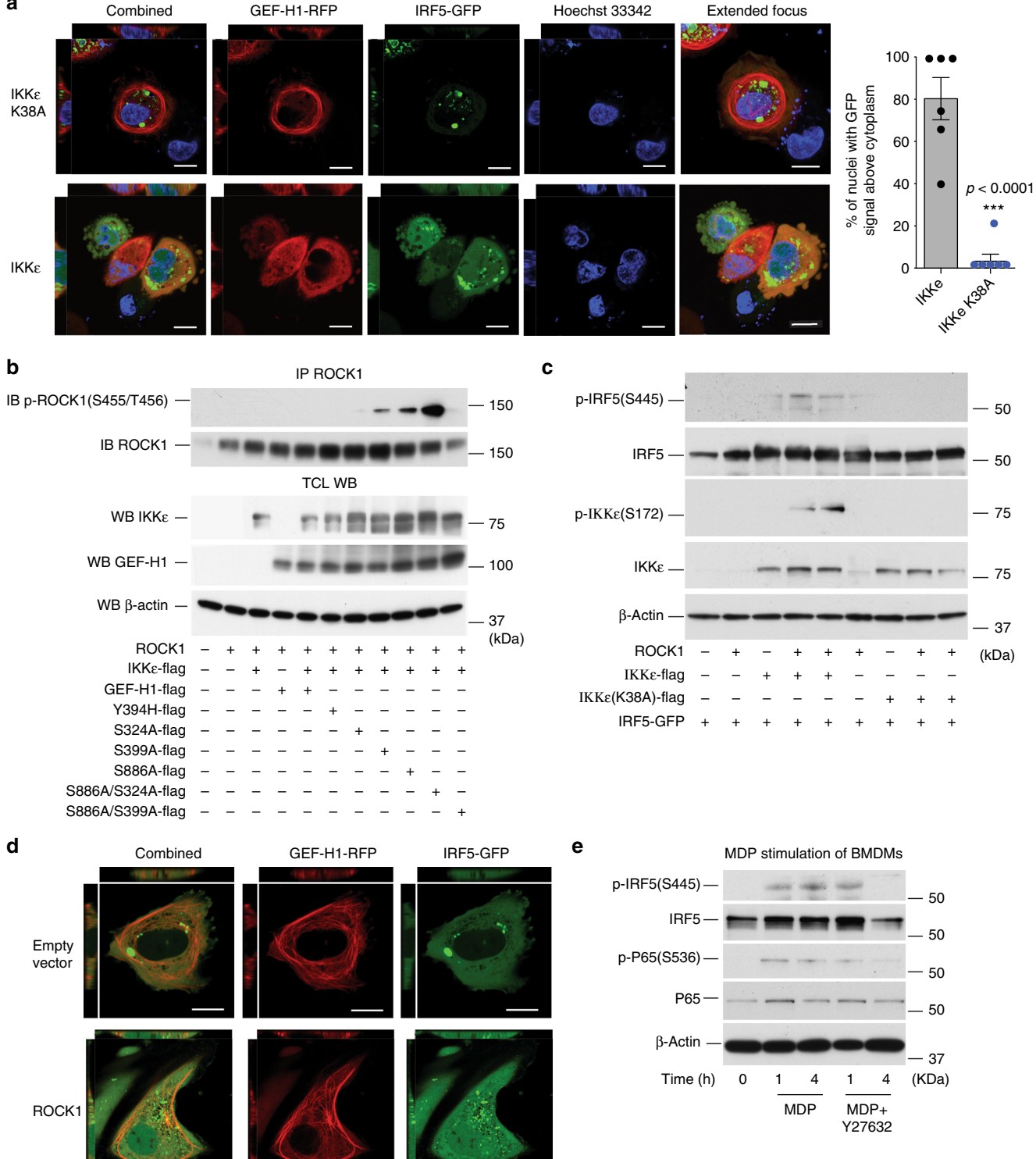

**Fig. 4** ROCK1/2 is required for IKKε and IRF5 activation. **a** Confocal microscopy analysis of primary human ARPE-19 cells 24 h after transfection with plasmids encoding GFP-tagged IRF5 and RFP-tagged GEF-H1 in presence of IKKε or IKKε (K38A) variant expressing constructs. Nuclear DNA was labeled using Hoechst 33342. Image acquisition was carried out with NIS-Elements imaging software (Nikon) followed by analysis by Volocity (PerkinElmer) to quantify the percentage of nuclei with GFP signal above cytoplasm in either IKKε or IKKε (K38A) transfected cells. Error bars indicate mean±SEM. Statistical significance was calculated using Student's *t*-test ***$P < 0.0001$ ($n = 6$). Scale bars, 10 μm. **b** Immunoblot analysis of ROCK1 phosphorylation after immunoprecipitation with anti-ROCK1, in HEK293T cells co-transfected with ROCK1 and IKKε plasmids in presence or absence of WT or GEF-H1 variants. **c** Western blot analysis of IRF5 and IKKε phosphorylation in response to WT or variant (K38A) IKKε expression in absence or presence of ROCK1 in HEK293T cells. **d** Confocal microscopy of ARPE-19 cells that were co-transfected with GFP-tagged IRF5 and RFP-tagged GEF-H1 plasmids in presence or absence of ROCK1 vector. Scale bars, 10 μm. **e** Immunoblot analysis of IRF5 and p65 phosphorylation in BMDMs from WT mice after stimulation with 5 μg ml$^{-1}$ of *N*-glycolyl-MDP with or without 20 μM of Y27632 (ROCK1/2 inhibitor). Source data are provided as a Source Data file

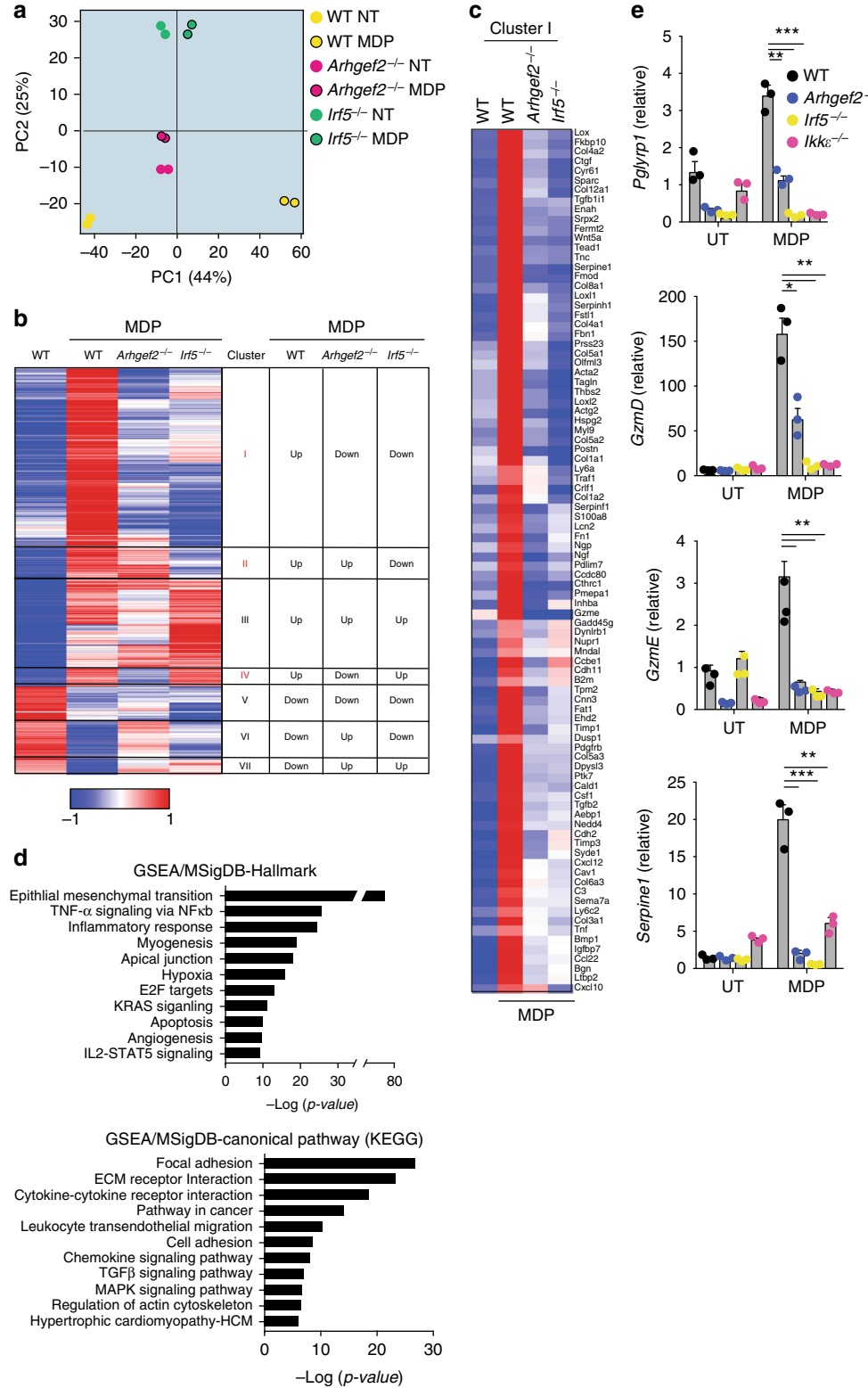

activates GEF-H1 or preventing binding of a kinase that inactivates GEF-H1 by phosphorylation of S886. Phosphorylation of S886 is specifically required for the binding of GEF-H1 to microtubules and dephosphorylation of this residue is required for the release and activation of GEF-H1[33,34]. However, we identified S324 as an additional Serine that controls GEF-H1 function. Preventing phosphorylation of S324 and S886 together enhanced GEF-H1 function significantly, dramatically increasing

the binding of IKKε and phosphorylation of ROCK1 as well as IRF5. In contrast, targeting S399 and S886 together in GEF-H1 prevented IKKε phosphorylation. Thus, we identified three Serine residues that are critical for the function of the GEF-H1-IKKε-IRF5 innate immune activation pathway. Targeting either S324 or S399 in addition to S886 of GEF-H1 maybe a strategy to enhance or limit immune responses that require GEF-H1. Our data also indicate that further analysis of GEF-H1 for functional motifs

**Fig. 5** GEF-H1 and IRF5 control transcriptional programs initiated by MDP. **a** PCA analysis of variant genes in which input samples are clustered in non-treated (open) and *N*-glycolyl-MDP stimulated (closed) BMDMs from WT (yellow circles), *Arhgef2*$^{-/-}$ (pink circles), and *Irf5*$^{-/-}$ (green circles) mice. **b** Heat map showing the differential expressed genes (DEG) upon *N*-glycolyl-MDP stimulation between BMDMs from WT, *Arhgef2*$^{-/-}$ and *Irf5*$^{-/-}$ mic. Scale represents Median centered log2FPKM. **c** Heat map representation of Cluster I shows the *N*-glycolyl-MDP regulated genes in untreated WT and *N*-glycolyl-MDP treated WT, *Arhgef2*$^{-/-}$ or *Irf5*$^{-/-}$ mice macrophages. Color scales represent **d** GSEA/MSigDB analysis showing significant pathways in Hallmark and Canonical pathway (KEGG) categories. DEGs were identified using an FDR cutoff <0.05 and a fold change cutoff >2 by Cuffdiff v1.06 in DNAnexus **e**, Gene expression analysis by qRT-PCR of *Pglyrp1*, *GzmE*, *GzmD*, and *Serpine-1* in BMDMs derived from WT (black square), *Arhgef2*$^{-/-}$ (blue square), *Irf5*$^{-/-}$ (yellow square), and *Ikkε*$^{-/-}$ (pink square), mice after 18 h stimulation with *N*-glycolyl-MDP or untreated control (UT). The data are presented as the mean±SEM. Statistical significance was tested with Student's *t*-test *$P < 0.05$, **$P < 0.001$, ***$P < 0.0001$ ($n = 3$). Source data are provided as a Source Data file. RNA sequencing data that support the findings of this study have been deposited in GEO with the accession codes GSE126749. http://www.ncbi.nlm.nih.gov/geo/query/acc.cgi?acc=GSE126749

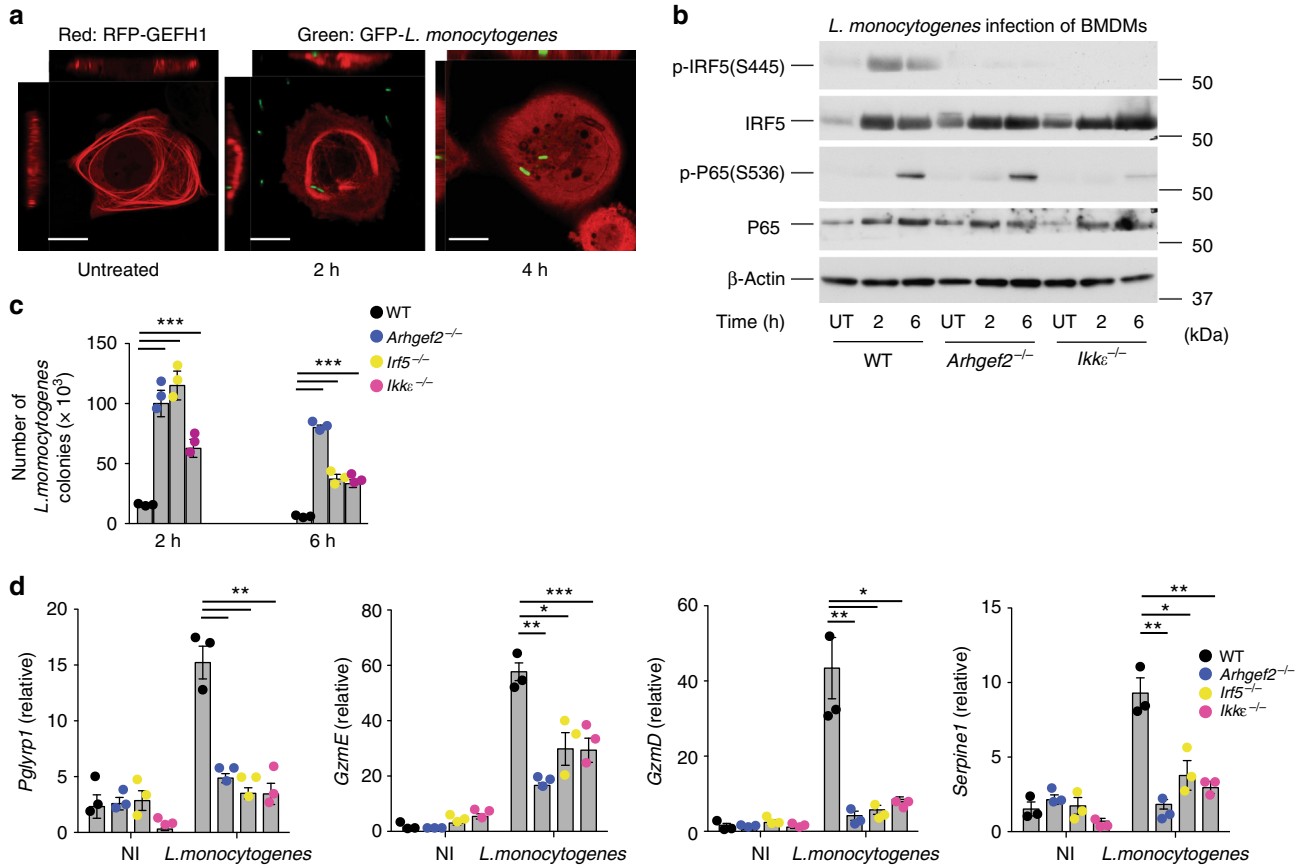

**Fig. 6** GEF-H1-IRF5 signaling controls host defense against *L. monocytogenes*. **a** Confocal microscopic analysis of subcellular localization of GEF-H1 in ARPE-19 before and during infection with *L. monocytogenes*. Scale bars, 10 μm. **b** Immunoblot analysis of *N*-glycolyl-MDP induced IRF5 and p65 phosphorylation in BMDMs isolated from WT, *Arhgef2*$^{-/-}$ and *Ikkε*$^{-/-}$ mice. **c** Gentamycin protection assay for the assessment of intracellular *L. monocytogenes* in BMDMs isolated from WT (black square), *Arhgef2*$^{-/-}$ (blue square), *Irf5*$^{-/-}$ (yellow square), and *Ikkε*$^{-/-}$ mice. **d** Gene expression analysis by qRT-PCR of *Pglyrp1*, *GzmE*, *GzmD*, and *Serpine1* in BMDMs derived from WT (black square), *Arhgef2*$^{-/-}$ (blue square), *Irf5*$^{-/-}$ (yellow square), and *Ikkε*$^{-/-}$ mice infected with *L. monocytogenes* or non-infected (NI). Error bars indicate mean±SEM. Statistical significance was calculated using Student's two-tailed *t* test **$P < 0.001$ ***$P < 0.0001$ ($n = 3$). Source data are provided as a Source Data file

should be carried out in mutants that prevent inactivation of GEF-H1 through phosphorylation of S886.

We found that ROCK1/2 may be required for IKKε phosphorylation by GEF-H1. The phosphorylation of IKKβ by RhoA GTPase function for the activation of NF-κB has been suggested in the TGF-β pathway, although in those experiments the responsible GEF had not been identified[35]. It will need to be determined whether ROCK1 or ROCK2 have cell specific function in activating IRF5 as the ROCK inhibitor used in these experiments can inhibit both ROCK kinases. Manipulation of the cytoskeleton by Rho GTPases can activate the NOD-Like receptor signaling pathway[36] and it needs to be determined whether

GEF-H1 can be activated through *L. monocytogenes* effectors that control the cytoskeleton and impact the microtubule network[37].

Surprisingly, GEF-H1-mediated IRF5 phosphorylation by MDP occurred in the absence of NOD2, a recognized sensor for MDP[38]. However, GEF-H1 and NOD2 both are required for the activation of NF-κB, indicating that NOD2 may specifically mediate NF-κB-p65 in response to MDP[39]. Although the role of GEF-H1 in the activation of IKKε was specific for the recognition of MDP, we found that IKKε was also required for TLR4-mediated IRF5 phosphorylation that occurs GEF-H1 independent. This indicates that IKKε has a broader unrecognized function in pathways that invoke IRF5-dependent transcriptional

responses. The specific signaling components that activate IKKε in the TLR pathway for the phosphorylation of IRF5 will need to be identified.

The GEF-H1-IKKε-IRF5 signaling axis was necessary for host defense against the enteroinvasive pathogen *L. monocytogenes*, that leads to a systemic bacterial infection which causes miscarriage in pregnant women, meningitis in neonates as well as the elderly and is often fatal to immunocompromised individuals. Furthermore, *Arhgef2*[−/−] macrophages were unable to significantly reduce the bacterial load after uptake, indicating that GEF-H1-mediated defense functions were required for the elimination of intracellular pathogens. *Arhgef2*[−/−] macrophages responded with NF-κB-p65 phosphorylation to *L. monocytogenes* uptake. This may be due to the activation of the STING pathway by microbial nucleic acids during *L. monocytogenes* infection[40]. Nevertheless, IKKε was of central importance also for the activation of NF-κB-p65 and IRF5 in these experiments. GEF-H1, IKKε, and IRF5 were specifically required for the induction of *Pglyrp1* in response to MDP and *L. monocytogenes* infection. Pglyrps participate in maintaining normal bacterial flora in the gut[41,42] and are critically involved in regulating inflammatory responses induced by bacteria together with NLRs[41]. However, the precise MDP receptor that activates GEF-H1 induced immune regulation to control innate and cell autonomous responses to commensal and pathogenic microbiota will need to be identified.

Altogether our data indicate that GEF-H1 can promote cell intrinsic innate and cell autonomous immunity by assembling a signalosome that allows the initiation of IRF5-dependent antibacterial transcriptional programs. The GEF-H1-IKKε-IRF5 host defense pathway is essential for the detection of peptidoglycans and enables host defense responses to cope with intracellular pathogens such as *L. monocytogenes*. In this pathway, GEF-H1 has an essential role for the unique activation of IKKε that has a specific function as an upstream IKKα/β and IRF5 kinase.

## Methods
**Cells lines and bone marrow-derived macrophages culture**. HEK293T and ARPE-19 cells were purchased from American Type Culture Collection. HEK293T were grown in DMEM and ARPE-19 in DMEM/F12 Medium, supplemented with 10% fetal bovine serum and 0.5% penicillin/streptomycin, Bone marrow-derived macrophages (BMDMs) cells were generated by flushing bone marrow cells from femurs and tibia of WT or indicated knockout (KO) mice, depleting red blood cells using ACK lysis buffer, and resuspending cell in complete DMEM media supplemented with 10% FBS, 0.5% penicillin/streptomycin mixture, and 20 ng ml[−1] M-CSF. Cells were maintained in culture at 37 °C, 5%CO$_2$ for 6 days before experimentation. BMDMs were stimulated in FBS free DMEM with *N*-Glycolyl-MDP (5 μg ml[−1]) (purchased from Invivogen; Cat# tlrl-gmdp), LPS-EK (100 ng ml[−1]; LPS from *E. coli* K12; Invivogen) or ROCK1 inhibitor Y27632 (20 μM; Abcam) after at least 2 h of serum starvation. Immortalized WT and *Ripk2* deficient macrophages cell lines were cultured in complete DMEM media supplemented with 10% FBS, 0.5% penicillin/streptomycin.

**Mice**. *Arhgef2*[−/−] mice were generated as previously described[3]. C57BL/6 WT (Wild-type), *Irf5*[−/−], *Nod2*[−/−], and *Ikkε*[−/−] animals were obtained from Jackson Laboratory (Bar harbor, ME). All animals were bred and housed in a pathogen-free animal facility according to institutional guidelines. All experiments were carried out on sex-matched mice at 8–12 weeks old with protocols approved by the subcommittee on Research Animal Care at the Massachusetts General Hospital and Harvard Medical School.

**Plasmids**. FLAG-tagged GEF-H1 plasmid (Human pCMV6-Entry-GEF-H1) vector was purchased from OriGene. pcDNA3-huIKKε-flag, pcDNA3-huIKKε (K38A)-flag, and pcDNA3.1-huIKKβ-HA plasmids were obtained from Addgene. pCMVtag2c-huNOD2-Flag was gift from Dr. Ramnik J. Xavier. The plasmids pCMV-huROCK1, pcDNA3-huRipK2-HA, and RFP-tagged GEF-H1 were previously described[2,3,39]. pcDNA3-huIRF5-GFP was kindly provided by Dr. Nancy C. Reich Marshall. GEF-H1 and GEF-H1-RFP variants (S324A Y394A, S399A, and S886A), were generated using the Quikchange Site-Directed Mutagenesis Kit (Stratagene) according to the manufacturer's instructions.

**Antibodies and western blotting**. The following antibodies were used in this study: rabbit antibodies against GEF-H1 (ab155785; 1/2000 dilution), phospho-GEF-H1 (S885(S886), ab94348; 1/2000 dilution), IRF5 (ab21689; 1/1000 dilution), Phospho-ROCK1(T455 /S456, ab203273; 1/1000 dilution), and ROCK1 (EP786Y, ab45171; 1/1000 dilution) were from Abcam. The antibody against phospho-Ser445 IRF5 used at 1/2000 dilution was generated by immunizing rabbits with a synthetic peptide (IRLQIpS445NPDLC; NeoBiolab, MA. USA). Phospho-p65-NFκB (Ser536, 93H1; 1/1000 dilution), p65-NFκB (D14E12; 1/1000 dilution), phospho-IRF3 (Ser396, 4D4G; 1/1000 dilution), IRF3 (D83B9; 1/1000 dilution), Phospho-IKKα/β (Ser176/180, 16A6; 1/1000 dilution), IKKβ (D30C6; 1/1000 dilution), Phospho-IKKε (Ser172, D1B7; 1/1000 dilution), IKKε (2690; 1/1000 dilution), β-actin (8H10D10; 1/10000 dilution), and anti-Lamin A/C (4C11; 1/1000 dilution) antibodies were purchased from Cell Signaling Technology. Anti-FLAG (F7425; 1/3000 dilution) and anti-HA (H9658; 1/3000 dilution) antibodies were obtained from Sigma. Whole-cell extracts were obtained by harvesting cells with lysis buffer (1% NP-40, 20 mM Tris-HCl (pH 7.4), 150 mM NaCl, 2 mM EDTA, 2 mM EGTA, 4 mM Na$_3$VO$_4$, and 40 mM NaF) containing protease and phosphatase inhibitors tablets (Roche). Western blotting was performed using standard protocols for SDS-PAGE and wet transfer onto PVDF membranes. Primary antibodies were diluted in blocking buffer (BSA 5% + 1X TBST) and incubated overnight at 4 °C. Secondary anti mouse (NA931V, GE Healthcare) or rabbit (NA934V, GE Healthcare) HRP were used at 1/5000 and incubated for 1 h at RT. For IP we used true blot anti-rabbit (ROCKLAND; 18-8816-33) or ULTRA anti-mouse (ROCKLAND; 18-8817-33) HRP at 1/4000. The bands were visualized by enhanced chemiluminescence (Western Lightning Plus [PerkinElmer] or SuperSignal West Femto [Thermo Fisher Scientific]) and exposure on film.

**Immunoprecipitation and Subcellular fractionation**. HEK293T cells were transfected with indicated plasmids using Lipofectamine 3000 (Invitrogen) and Amaxa Mouse Nucleofector® Kit (Lonza, Cat#VPA-1009) used for Macrophages transfection according to the manufacturer's protocol. The cells were lysed on ice for 20 min, in the same lysis buffer used above to harvest cell lysate for immunoblot. Cell debris was pelleted by centrifugation and the supernatant was then incubated for 30 min at 4 °C with protein G plus agarose (Pierce Thermo Scientific, Rockford, IL) Precleared lysates were incubated at 4 °C overnight with immunoprecipitation antibodies. The protein G agarose beads were then added, and the incubation continued for 4 h. Following extensive washes with the same lysis buffer, the agarose beads were mixed with 1 × SDS sample buffer and boiled for 5 min prior to immunoblotting analysis.

Nuclear and cytoplasmic extracts from BMDMs were prepared using Buffer A (10 mM HEPES, pH7.9; 1.5 mM MgCl2; 10 mM KCl; 0.1 mM EDTA; 0.1 mM EGTA; 1 mM DTT; 0.3 mM Na3VO4 + protease inhibitors tablet [Roche]) and Buffer C (20 mM HEPES, pH7.9; 1.5 mM MgCl2; 1 mM EDTA; 1 mM EGTA; 1 mM DTT; 0.3 mM Na3VO4; 0.4 M NaCl + protease inhibitors tablet [Roche]). In brief, cells were collected by scraping and centrifugation 5 min at 3000 rpm, 400 μl of Buffer A added to the pellet and after incubation 15 min on ice, 50 μl of 10% NP-40 was added and the supernatant collected (cytosolic fraction) after centrifugation at 15,000 rpm for 30 s. A volume 50 μl of Buffer C added to the pellet, vigorously rocked at 4 °C for 15 min and centrifuged 5 min at 15,000 rpm. The collected supernatant represents the nuclear extracts. Equal amounts of nuclear protein were loaded in each lane and separated on a 4–20% Tirs-Gly NuPAGE® gel (Invitrogen), then transferred to a PVDF membrane. Membranes were probed with anti IRF5 antibody or anti-Lamin A/C. Uncropped and unprocessed scans of the most important blots are provided as a Source Data file

**Real-time quantitative-PCR**. Total RNA from BMDMs was isolated using RNeasy micro kit (Qiagen). cDNA was prepared from RNA using iScript cDNA synthesis kit (Bio-Rad). Real-time PCR was performed using SsoAdvanced™ Universal SYBR green Supermix (Bio-Rad). The gene expression was normalized to the expression of the gene encoding 18S. The primer sequences are provided in Supplementary Table 1.

**Bacterial killing assay in vitro**. BMDMs derived from WT, *Arhgef2*[−/−], *Irf5*[−/−], or *Ikkε*[−/−] mice were plated to 24-well plate at a density of 1 × 10$^5$ cells per well. Cells were infected with GFP-*Listeria monocytogenes* (generous gift from Dr. John Garber) at a multiplicity of infection (MOI) 1 for 1 h at 37 °C. BMDMs were washed with sterile HBSS and the extracellular bacteria was eliminated by incubation for 1 h with 100 μg ml[−1] gentamicin, Cells were collected (this was considered as time point 2 h) or incubated for another 4 h without gentamicin, which represent the 6 h time point. After washing with HBSS, BMDMs were disrupted for 15 min with 250 μL dH$_2$O. Intracellular bacteria were enumerated by serial dilution and spread on Brain Heart Infusion (BHI) agar plates with chloramphenicol.

For western blot assay, BMDMs were seeded at a density of 1 × 10$^6$ per well in 12-well plates. Cells were infected with *L. monocytogenes* at a MOI 1 and harvested at time point 2 h and 6 h for protein analysis with immunoblotting.

**RNAseq and GSEA analyses**. Total RNA was isolated from BMDMs derived from WT, *Arhgef2*[−/−], and *Irf5*[−/−] mice, using RNeasy Micro kit (Qiagen). Libraries were synthesized using Illumina TruSeq Stranded mRNA sample preparation kit from 500 ng of purified total RNA and indexed adapters according to the

manufacturer's protocol (Illumina). The final dsDNA libraries were quantified by Qubit fluorometer, Agilent Tapestation 2200, and RT-qPCR using the Kapa Biosystems library quantification kit according to manufacturer's protocols. Pooled libraries were subjected to 35-bp paired-end sequencing according to the manufacturer's protocol (Illumina NextSeq 500). Targeted sequencing depth was 30 million paired-end reads per sample. Blc2fastq2 Conversion software (Illumina) was used to generate de-multiplexed Fastq files.

Expression values were normalized as Fragments per Kilobase Million reads after correction for gene length (FPKM) in Cuffdiff version 1.06 in the DNAnexus analysis pipeline and filtered for genes that exhibited a statistically significant difference ($P < 0.01$) with a false discovery rate threshold of 0.05 and a biologically relevant change log-fold change >1. Samples were analyzed in the RNA-seq pipeline of Seqmonk for mRNAs for opposing strand specific and paired end libraries with merged transcriptome isoforms, correction for DNA contamination and log transformed resulting expression values in log2 FPKM. MDP induced mRNAs that were differentially regulated more that twofold (FDR threshold of 0.05) in the Cuffdiff analysis of WT BMDMs were imported into Seqmonk for per-probe normalized hierarchical clustering of mRNA transcription in control and $N$-Glycolyl-MDP stimulated WT and $Arhgef2^{-/-}$ BMDMs.

To generate a ranked gene list for GSEA analyses stranded reads were aligned and counted using STAR (2.5.2a)[43] in stranded union mode using Illumina's ENSEMBL iGenomes GRCm38 build and GRCm38.90 known gene annotations. Count level data were then analyzed using the edgeR Bioconductor package in R[44]. Filtered genes, expressed at >1 count per million (cpm) in at least two samples, were analyzed using the QLF functions comparing untreated and MDP-treated WT and $Arhgef2^{-/-}$, $Irf5^{-/-}$ BMDMs. All genes were ranked according to their –log10 transformed corrected $p$-value for differential up/down-regulation by MDP in WT versus $Arhgef2^{-/-}$ and $Irf5^{-/-}$ samples. Mouse genes were mapped to their human orthologs using HCOP (http://www.genenames.org/cgi-bin/hcop). The pre-ranked list was used to perform weighted GSEA using the GSEA java application (http://www.broad.mit.edu/gsea/) that uses the Molecular Signature Database (MSigDB)[45].

**Live cell imaging**. About 50,000 ARPE-19 cell were plated in Nunc™ Lab-Tek™ Chambered Coverglass (Cat.155383PK, Thermo Scientific) and transfected with 200 ng of each indicated plasmid using lipofectamine 3000.

For GFP tagged-*L. monocytogenes* experiment, the cells were transfected with RFP-tagged GEF-H1 and infected with a MOI 10. Live cells were imaged with a Nikon A1R-A1 confocal microscope. Image acquisition was carried out with NIS-Elements imaging software (Nikon) followed by analysis with Volocity (PerkinElmer)

**Statistical analysis**. Error bars indicate mean±SEM. All statistical significance was performed with GraphPad Prism Software (version6.01; GraphPad, San Diego, CA) using two-tailed *t*-test. Statistical significance was assumed at $p < 0.05$. All experiments were repeated at least two times.

**Reporting summary**. Further information on experimental design is available in the Nature Research Reporting Summary linked to this article.

## Data availability
RNA sequencing data that support the findings of this study have been deposited in GEO with the accession codes GSE126749. http://www.ncbi.nlm.nih.gov/geo/query/acc.cgi?acc=GSE126749.

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

## Acknowledgements

This work was supported by grants DK068181 (H.C.R.), AI113333 (H.C.R.), DK033506 (H.C.R.), and DK043351 (H.C.R.) from the National Institutes of Health.

## Author contributions

H.C.R. designed experiments; and Y.Z., R.Z., S.M.P, N.Y., and P.S. carried out experiments; Y.Z., R.Z., S.M.P and H.C.R. analyzed and interpreted data, and Y.Z., R.Z, S.M.P. and H.C.R. wrote the paper.

## Additional information

**Competing interests:** The authors declare no competing interests.

