## [Peer Review File · Nature Communications]

Reviewers' comments:

Reviewer #1 (Remarks to the Author):

Summary: The manuscript entitled "Microbial Recognition by GEF-H1 Controls IKK ϵ Mediated Activation of IRF5" by Zhao et al, describes a microtubule based recognition system for muramyl dipeptides for innate immune responses through the activation of IRF5. The guanine nucleotide exchange factor GEF-H1 is required for the activation of IKK α/β and IRF5. The GEF-H1 interacting kinase RIPK2 is required for IRF5 phosphorylation.

Comments:

Figure 1a: Loading for p65 is uneven. There appears to be less p65 in the KO lanes and some induction of p-65 phosphorylation in the Arhgef2 Kos. The western blot images should be quantified through the manuscript particularly when small differences in the magnitude of the signals are asserted.

1b. Cytoplasmic controls should be included for the localization study on IRF5 to track the total cellular amounts of IRF5. Does the cytoplasmic fraction similarly decrease in time with MDP stimulation?

1c. IRF5 coprecipitates with GEF-H1 under overexpression conditions in HEK293 cells. Can this association be seen with normally expressed proteins in macrophages? GEF-H1 associates with microtubules. Does IRF5 associate with MTs? Is the interaction with GEF-H1 indirect through MT or direct? Does association survive Nocodazole treatment?

Figure 2a: The authors observe enhanced IRF5 phosphorylation with both IKK ϵ and GEF-H1 are coexpressed in HEK cells. HEK cells express GEF-H1. Does the increase in IRF5 phosphorylation occur with overexpression of IKK ϵ with GEF-H1 is knockdown, ko'd by CRISPR or in the ARHGEF2 KO cells? Is the effect of GEF-H1 on assembling this MDP sensor dependent on its exchange activity? That is what is the behavior of the catalytically inactive GEF-H1 mutant?

2g: There is residual IRF5 phosphorylation in the Arhgef2 Kos.

An important step in RIPK2 activation is the Lys-63 linked polyubiquitination by XIAP/BIRC2 and BIRC3. Are there changes in RIPK2 ubiquitylation in the Arhgef2 KOS?

Fig3a is the diagram. Not 2a. There is no sequence alignment. Residues 298, 367 and 372 are all within the DH catalytic domain. Do these mutations alter the catalytic activity of the GEF-H1?

3c. The use of "pan S/T" antibodies is fraught with problems of lack of both sensitivity and specificity. Since GEF-H1 is a heavily phosphorylated protein, it is difficult to interpret these data.

3d. "Remarkably, the autophosphorylated GEF-H1 (S372A) variant was unable to bind IKK ϵ ." GEF-H1 is not a kinase and therefore cannot "autophosphorylate".

The authors state: "Together these data demonstrated that Serines in the pLxIS and the YPLxIS domains of GEF-H1 mediate the interaction with IKK ϵ and its activation". Short peptide sequences are not domains but rather motifs. Can pLxIS and or the YPLxIS synthetic peptide directly interact with IKK ϵ ? What do the authors propose is the mode of protein interaction between GEF-H1 and IKK ϵ ?

4a. Note that the subcellular localization of GEFH1 is "bundled" into a birds nest configuration. This effect has been well documented by G. Bokoch and others as a microtubule stabilizing effect. High levels of GEF-H1 create a pathological microtubule array. This effect could alter the results and the authors should comment on this.

4c. Phosphospecific antibodies are renowned to be non-specific since they are raised against short peptide sequences. ROCK1 should first be IP'd then probed with the pROCK antibody. Same for IRF5 and IKK ϵ (misspelled in 4e). What is the effect of a ROCK inhibitor on IKK ϵ phosphorylation? Again the role of GEF-H1 in activating ROCK and phosphorylating IRF5 should be examined with a catalytically inactive GEF-H1 mutant.

Fig.6 GEF-H1 regulates actin dynamics and tail retraction in granulocytes and macrophages and is part of a chemomotor apparatus in immune cells. To what extent does the failure to clear infection due to failure of ARFGEF2 KO immune cells to mobilize? GEF-H1 also has reported functions in T cells. To what extent does the immune suppressed state a reflection of the role of GEF-H1 in these other cellular types? Have the authors looked at MAC specific deletion of GEF-H1 to examine these innate immune responses more specifically?

General comments:

What is the receptor for MDP in this model?

Reviewer #2 (Remarks to the Author):

In this issue, Zhao et al demonstrate that IKK ϵ is compulsory for IRF5 phosphorylation under MDP dependant GEF-H1 stimulation.

This conclusion is supported by very elegant WB experiments, transcriptomic analysis and confocal microscopy performed either on over expression system in HEK and on BMDM from various KO mice.

The implication of IKK ϵ in GEF-H1 dependant IRF5 activation is novel, and will have a significant impact in the field of innate immunology.

Minor comments:

Fig1a: A positive control for pIRF3 would be required to conclude in a lack of activation.

Fig2b and line 119: the authors state: "IKKε^{-/-} macrophages were unable to respond to MDP stimulation with 118 the phosphorylation of S445 of IRF5 and S536 of p65, respectively (Figure 2b)".

The p-P65 (S536) results are not as clear as the pIRF5 (S445) one, especially given the heterogeneity for beta-actin and total p65 between WT and IKKε^{-/-} cells. Conclusion on IKKε dependant NFκB phosphorylation in this model would require further evidence.

Fig2c: A positive control for pTBK1 would be required to conclude in a lack of activation

Fig2a, d, e, f: could the authors clarify if the experiment are performed with or without MDP.

Line151 and 155: Could the autors clarify what AA stand for?

Line 151: Figure 3a instead of 2a

Line 180: pIRF5 is detectable with IKKε expression alone. Maybe not detectable could be replace by barely detectable?

Line196: could the authors explain why they decide to use the Arpe19 model for microscopy experiments?

Fig 4a: The authors state: "Upon co-expression of GEF-H1 together with IKKε or RipK2, IRF5 translocated to the nucleus (Figure 4a)". This would not be my interpretation of the representative figures for IKKε. Conclusion on IRF5 translocation would strongly benefit quantification of staining overlap with a nuclear co staining.

Line 213: (Figure 4e) instead of Figure 4d

Line 239: (Figure 5c) instead of figure 5b

Please, provide supplementary Tables with gene list from each cluster, as indicated in the manuscript (line 242 and 245)

Line 284: 6 hours instead of 4?

Reviewer #3 (Remarks to the Author):

Zhao et al., report a role for Guanine nucleotide exchange factor H1 in innate immune activation by microbial muramyl-dipeptides (MDP) in the manuscript entitled "Microbial Recognition by GEF-H1 Controls IKKε Mediated Activation of IRF5". Peptidoglycan is one of the main microbial ligands

recognized by the innate immune cells during infections with Gram-positive bacteria. However, the signaling components involved in this process are incompletely understood. Zhao et al., describe that GEF-H1 is required for the activation of IRF5 by MDP. GEF-H1 released from microtubules following MDP detection mediates the assembly of a signaling complex composed of RipK2, IKK ϵ , IKK α/β and IRF5, resulting in the activation of IKK ϵ and IRF5. This is a well-written manuscript with interesting findings. However, the manuscript has the following issues.

Major comments

This manuscript excessively relies on overexpression experiments in 293T cells to demonstrate GEF-H1/IKK ϵ /IRF5 interactions. The obvious caveat is that these results may or may not reflect endogenous interactions in immune cells. Therefore, all the overexpression data should be corroborated by IP studies on native/endogenous interactions in MDP stimulated or infected macrophages.

It is important to more thoroughly examine the role of GEF-H1 in innate immune responses of macrophages/dendritic cells to bacteria such as *Listeria* in addition to the studies in Fig 5. It is recommended that the authors examine the expression of additional immune and inflammatory genes at the RNA and protein levels in WT and GEF-H1-deficient macrophages during *Listeria* infection.

Similarly, the authors should perform *in vivo* studies utilizing GEF-H1-deficient mice to demonstrate if GEF-H1/IKK ϵ /IRF5 axis contribute to host defense responses and bacterial control during infections. This would substantially improve the significance of this study.

Additional comments

Fig 2b and 4f. The levels of total IRF5 itself were lower in KO and inhibitor treated lysates, which could explain the reduction in pIRF5 levels in the corresponding lanes.

In many instances the authors' claim were inconsistent with the data presented.

- Fig 1a. pIRF3 and pIRF5 were present at similar levels in WT cells. But authors claim that MDP activated IRF5 but not IRF3.

- Fig 2g. pIRF5 levels were lower in LPS-treated GEF-H1 KO cells compared to WT cells . But the authors claim that GEF-H1 was specifically required during MDP induced IRF5 activation.

-Fig 3g. S372A mutant appears to be defective for IRF5 recruitment.

It is not clear how the authors have determined that GEF-H1 localized to microtubules. Similarly, the confocal data is not convincing that GEF-H1 was released from microtubules. This should be corroborated using additional approaches as well as in macrophages upon MDP stimulation.

Fig 1f. Does GEF-H1 pull down NOD2 in the absence of IKKε?

Is GEF-H1 constitutively phosphorylated at S372 in macrophages?

GEF-H1 Controls IKK ϵ Mediated Activation of IRF5 for Host Defense

Dear referees, we have extensively revised the manuscript to address the concerns

The manuscript demonstrates that GEF-H1 is activated by dephosphorylation in the presence of MDP and *L. monocytogenes* to interact with and activate IKK ϵ and IRF5 to initiate host defenses in macrophages. In the revised manuscript we were able to:

1. Demonstrate the interaction of GEF-H1 with IRF5 in primary macrophages (New Figure 1d).
2. Include control experiment demonstrating that MDP failed to induce TBK-1 and IRF3 phosphorylation in macrophages in contrast to cyclic-di-GMP induced STING activation (New Supplemental Figure S1).
3. Extend the mutational analysis of GEF-H1 motifs that mediate interaction and innate immune activation. In double mutants we have removed S886 to prevent hyper-phosphorylation and conformational inactivation of GEF-H1 allowing to more specifically determine the role of S324 and S399 for IRF5 and IKK ϵ activation (New Figures 3a, 3e and 3f).
4. Integrate new confocal image analysis that demonstrates that IKK ϵ but not a kinase deficient IKK ϵ variant can induce release of GEF-H1 from the microtubule network and nuclear translocation of IRF5 (New Figure 4a).
5. Include new experiments in which we immunoprecipitated ROCK1 first and demonstrate the phosphorylation of Rock1 in the presence of IKK ϵ and active GEF-H (New Figure 4b). We also included experiments in which we show that ROCK1 induces nuclear translocation of IRF5. (New Figure 5d).
6. Expand the experiments that demonstrate that IRF5 activation by *L. monocytogenes* is dependent on GEF-H1 and IKK ϵ . New experiment show that the regulation of genes that are specifically induced by MDP in a GEF-H1 and IRF5 dependent manner in macrophages, also require GEF-H1, IKK ϵ and IRF5 during *L. monocytogenes* infection for transcriptional induction (New Figure 6d).

New and revised figures include Figure 1a, 1b, 1d, 2g, 3a, 3e, 3f, 4a, 4b, 4c, 4d, 6b, 6d, and Supplemental figure S1.

Reviewer #1 (Remarks to the Author):

Summary: The manuscript entitled "Microbial Recognition by GEF-H1 Controls IKK ϵ Mediated Activation of IRF5" by Zhao et al, describes a microtubule based recognition system for muramyl dipeptides for innate immune responses through the activation of IRF5. The guanine nucleotide exchange factor GEF-H1 is required for the activation of IKK α/β and IRF5. The GEF-H1 interacting kinase RIPK2 is required for IRF5 phosphorylation. Figure 1a: Loading for p65 is uneven. There appears to be less p65 in the KO lanes and some induction of p-65 phosphorylation in the Arhgef2 Kos. The western blot images should be quantified through the manuscript particularly when small differences in the magnitude of the signals are asserted.

Answer: We have revised these figures included new data, additional controls and included quantifications were small differences were observed.

Critique 1: 1b. Cytoplasmic controls should be included for the localization study on IRF5 to track the total cellular amounts of IRF5. Does the cytoplasmic fraction similarly decrease in time with MDP stimulation?

Answer: Total IRF5 expression is induced during MDP stimulation and increases over the observed time period as shown in Figure 1a. Of this a sub-fraction of IRF5 is phosphorylated and targeted to the nucleus.

Critique 2: 1c. IRF5 coprecipitates with GEF-H1 under overexpression conditions in HEK293T cells. Can this association be seen with normally expressed proteins in macrophages? GEF-H1 associates with microtubules. Does IRF5 associate with MTs? Is the interaction with GEF-H1 indirect through MT or direct? Does association survive Nocodazole treatment?

Answer: We were able to demonstrate co-immunoprecipitation of GEF-H1 and IRF5 in WT but not *Arhgef2*^{-/-} macrophages in Figure 1c. We did not observe microtubule association of IRF5 as shown in Figure 4a and 4d. The current model proposed requires the activation and release of GEF-H1 from microtubules to allow IKK ϵ and IRF5 activation. Nocodazole stabilizes microtubules and thus prevents GEF-H1 activation in the RLR pathway¹. In a different manuscript under consideration by Nature Communication (NCOMMS-18-19571A-Z) we have independently analyzed immune cell activation by GEF-H1 in response to microtubule destabilizers that activate GEFH1 and IRF5 versus microtubule stabilizers that cannot activate GEFH1 dependent signaling in the context of anti-tumor immune responses.

Critique 3: Figure 2a: The authors observe enhanced IRF5 phosphorylation with both IKK ϵ and GEF-H1 are co-expressed in HEK cells. HEK cells express GEF-H1. Does the increase in IRF5 phosphorylation occur with overexpression of IKK ϵ with GEF-H1 is knockdown, ko'd by CRISPR or in the ARHGEF2 KO cells? Is the effect of GEF-H1 on assembling this MDP sensor dependent on its exchange activity? That is what is the behavior of the catalytically inactive GEF-H1 mutant?

Answer: We show that macrophages derived from IKK ϵ as well as GEF-H1 deficient mice cannot efficiently activate IRF5 in response to MDP and *L. monocytogenes* (Figure 2b and Figure 6b). These experiments establish the requirement of GEF-H1 and IKK ϵ for IRF5 activation in this pathway. We further integrated new confocal image analysis that demonstrates that IKK ϵ but not a kinase deficient IKK ϵ variant can induce release of GEF-H1 from the microtubule network and nuclear translocation of IRF5 (Figure 4a).

Critique 4: 2g: There is residual IRF5 phosphorylation in the *Arhgef2* Kos.

An important step in RIPK2 activation is the Lys-63 linked polyubiquitination by XIAP/BIRC2 and BIRC3. Are there changes in RIPK2 ubiquitylation in the *Arhgef2* KOS?

Answer: We have not endeavored into this distinct area of investigation but this is a possible direction for future experiments. *Arhgef2* KOs still express IKK β it is possible that some activation of IKK β through the NF- κ B pathway contributes to the residual IRF5 activation. However, IKK ϵ KOs are unable to activate IRF5 in response to MDP.

Critique 5: Fig3a is the diagram. Not 2a. There is no sequence alignment. Residues 298, 367 and 372 are all within the DH catalytic domain. Do these mutations alter the catalytic activity of the GEF-H1?

Answer: We have corrected this omission (New Figure 3a). The new model of GEF-H1 that emerges from the analysis of double mutants is that phosphorylation of S324 and S886 prevents access to catalytic activity of GEF-H1 for ROCK1 activation (Figures 4b, 4c, 4d and 4e). The catalytic deficient mutant Y394H fails to induce ROCK1 activation in the presence of IKK ϵ . Further, ROCK1 can activate IKK ϵ and IRF5 phosphorylation and is required for IRF5 activation macrophages (Figure 4c).

Critique 6: 3c. The use of “pan S/T” antibodies is fraught with problems of lack of both sensitivity and specificity. Since GEF-H1 is a heavily phosphorylated protein, it is difficult to interpret these data.

Answer: The data was included to demonstrate that there are additional phosphorylation events that occur upon IKK ϵ but not IKK β co-expression that need to be characterized in future experiments (see comparison in Figure 3c).

Critique 7: 3d. “Remarkably, the autophosphorylated GEF-H1 (S372A) variant was unable to bind IKK ϵ .” GEF-H1 is not a kinase and therefore cannot “autophosphorylate”.

Answer: We have modified this in describing this variant as ‘hyper-phosphorylated’.

Critique 8: The authors state: “Together these data demonstrated that Serines in the pLxIS and the YPLxIS domains of GEF-H1 mediate the interaction with IKK ϵ and its activation”. Short peptide sequences are not domains but rather motifs. Can pLxIS and or the YPLxIS synthetic peptide directly interact with IKK ϵ ? What do the authors propose is the mode of protein interaction between GEF-H1 and IKK ϵ ?

Answer: The revised data provided in Figure 3d, 3e, and 3f show that the interaction of GEF-H1 with IKK ϵ and IRF5 is regulated by dephosphorylation events in these motifs.

Dephosphorylation of S886 is a prerequisite for allowing interaction with IKK ϵ . S324 is further critical as its removal dramatically enhances binding to IKK ϵ and IRF5. This dominant negative S324A/S886A variant of GEF-H1 also remarkably increases phosphorylation of ROCK1 (New Figure 4b). S399 is a critical residue that controls phosphorylation at S886 and thus the inactivation of GEF-H1 (Figure 3b). The identification of the critical serines in GEF-H1 was possible through experiments with double mutants of GEF-H1 that cannot be inactivated by phosphorylation of S886 (Figure 3e, 3f and Figure 4b).

Critique 9: 4a. Note that the subcellular localization of GEF-H1 is “bundled” into a birds nest configuration. This effect has been well documented by G. Bokoch and others as a microtubule stabilizing effect. High levels of GEF-H1 create a pathological microtubule array. This effect could alter the results and the authors should comment on this.

Answer: We have included this important point. Of note continued stimulation of the pathway through IKK ϵ overexpression (Figure 4a) or *L. monocytogenes* infection (Figure 6a) prevented the formation of MT bundles that occur during overexpression of GEF-H1.

Critique 10: 4c. Phosphospecific antibodies are renowned to be non-specific since they are raised against short peptide sequences. ROCK1 should first be IP'd then probed with the pROCK antibody. Same for IRF5 and IKK ϵ (misspelled in 4e). What is the effect of a ROCK inhibitor on IKK ϵ phosphorylation? Again the role of GEF-H1 in activating ROCK and phosphorylating IRF5 should be examined with a catalytically inactive GEF-H1 mutant.

Answer: We have carried the experiments accordingly (New Figure 4a). We immunoprecipitated ROCK1 from HEK293T cells that were transfected with ROCK1, IKK ϵ , and WT and variants of GEF-H1 to determine ROCK1 phosphorylation at S455/T456. Remarkably, the GEF-H1 double mutant S324/S886A allowed the highest phosphorylation of ROCK1 while the S399A/S886A variant failed to induce significant ROCK1 activation (Figure 4b). The GEF-H1 variants S324A, S399A and S886A were able to confer some ROCK1 phosphorylation in the presence of IKK ϵ when compared to WT GEF-H1 or the exchange deficient mutant GEF-

H1Y394H (Figure 4b). These experiments indicated that dephosphorylation of GEF-H1 allows the activation of ROCK1 as a requirement for IRF5 activation. Indeed, ROCK1 mediated phosphorylation of IKK ϵ and IRF5 in the presence of IKK ϵ but not inactive IKK ϵ K38A in HEK293T cells (Figure 4c).

Critique 11: Fig.6 GEF-H1 regulates actin dynamics and tail retraction in granulocytes and macrophages and is part of a chemomotor apparatus in immune cells. To what extent does the failure to clear infection due to failure of ARFGEF2 KO immune cells to mobilize? GEF-H1 also has reported functions in T cells. To what extent does the immune suppressed state a reflection of the role of GEF-H1 in these other cellular types? Have the authors looked at MAC specific deletion of GEF-H1 to examine these innate immune responses more specifically?

Answer: These are important Critiques and there is the possibility that additional functions of GEF-H1 critically control immune responses but that will require extensive further investigations and new mouse model systems. Baseline distribution of immune cell subsets is not impacted in GEF-H1 deficient mice ¹ but whether microbial challenges will impact the complex interaction of macrophage, dendritic cell and T cells subsets in forming an immune response is the subject of further investigations.

Critique 12: *General comments: What is the receptor for MDP in this model?*

Answer: Our results indicate that the true MDP receptor needs to be identified. We are pursuing the identification of these receptors, which is hampered by the lack of specific tools and functional antibodies, which we are creating.

Reviewer #2 (Remarks to the Author):

In this issue, Zhao et al demonstrate that IKK ϵ is compulsory for IRF5 phosphorylation under MDP dependant GEF-H1 stimulation. This conclusion is supported by very elegant WB experiments, transcriptomic analysis and confocal microscopy performed either on over expression system in HEK and on BMDM from various KO mice. The implication of IKK ϵ in GEF-H1 dependant IRF5 activation is novel, and will have a significant impact in the field of innate immunology.

Minor comments: Fig1a: A positive control for pIRF3 would be required to conclude in a lack of activation.

Fig2b and line 119: the authors state:“IKK ϵ -/- macrophages were unable to respond to MDP stimulation with 118 the phosphorylation of S445 of IRF5 and S536 of p65, respectively. The p-P65 (S536) results are not as clear as the pIRF5 (S445) one, especially given the heterogeneity for beta-actin and total p65 between WT and IKK ϵ -/- cells. Conclusion on IKK ϵ dependant NF-kB phosphorylation in this model would require further evidence.

Fig2c: A positive control for pTBK1 would be required to conclude in a lack of activation

Answer: We have included new Figure 1a and 1b that demonstrate IRF5 phosphorylation in response to MDP in WT and NOD2 deficient but not GEF-H1 deficient mice. We have included control experiments that demonstrate that MDP fails to induce IRF3 and TBK1 phosphorylation compared to experiments using c-diGMP as a STING dependent activator (Supplemental Figure S1).

Critique 1: Fig 2a, d, e, f: could the authors clarify if the experiment are performed with or without MDP.

Answer: We have emphasized this information in the figure directly and the figure legend. Experiments in Figure 2b, 2c, 2e, 2f, 2g were carried out with macrophages from WT, RipK2, GEF-H1 and IKK ϵ deficient mice stimulated with MDP.

Critique 2: Line 151 and 155: Could the authors clarify what AA stand for?

Answer: We clarified the motifs involved and re-numbered amino acids according to isoform 1 of GEF-H1 (NP_001155855.1).

Critique 3: Line 151: Figure 3a instead of 2a

Answer : This was corrected

Critique 4: Line 180: pIRF5 is detectable with IKK ϵ expression alone. Maybe not detectable could be replaced by barely detectable?

Answer: We have made this change.

Critique 5: Line 196: could the authors explain why they decide to use the Arpe19 model for microscopy experiments?

Answer: ARPE19 cells are primary human ocular epithelial cells. They were chosen over tumor derived cell lines with uncertain number of chromosomes and gene duplications. These cells also have a well-developed microtubule systems that can be imaged and are transfectable.

Critique 6: Fig 4a: The authors state: "Upon co-expression of GEF-H1 together with IKK ϵ or RipK2, IRF5 translocated to the nucleus (Figure 4a)". This would not be my interpretation of the representative figures for IKK ϵ . Conclusion on IRF5 translocation would strongly benefit quantification of staining overlap with a nuclear co staining.

Answer: We integrated new confocal analysis image analysis that demonstrates that IKK ϵ but not a kinase deficient IKK ϵ variant can induce release of GEF-H1 from the microtubule network and nuclear translocation of IRF5 (Figure 4a).

Critique 7: Line 213: (Figure 4e) instead of Figure 4d, Line 239: (Figure 5c) instead of figure 5b. Please, provide supplementary Tables with gene list from each cluster, as indicated in the manuscript (line 242 and 245)

Line 284: 6 hours instead of 4?

Answer: We have corrected these omissions and included a Supplemental data table with this information.

Reviewer #3 (Remarks to the Author):

Zhao et al., report a role for Guanine nucleotide exchange factor H1 in innate immune activation by microbial muramyl-dipeptides (MDP) in the manuscript entitled "Microbial Recognition by GEF-H1 Controls IKK ϵ Mediated Activation of IRF5". Peptidoglycan is one of the main microbial ligands recognized by the innate immune cells during infections with Gram-positive bacteria. However, the signaling components involved in this process are incompletely understood. Zhao et al., describe that GEF-H1 is required for the activation of IRF5 by MDP. GEF-H1 released from microtubules following MDP detection mediates the assembly of a signaling complex composed of RipK2, IKK ϵ , IKK α/β and IRF5, resulting in the activation of IKK ϵ and IRF5. This is

a well-written manuscript with interesting findings. However, the manuscript has the following issues.

Major comments

This manuscript excessively relies on overexpression experiments in 293T cells to demonstrate GEF-H1/IKK ϵ /IRF5 interactions. The obvious caveat is that these results may or may not reflect endogenous interactions in immune cells. Therefore, all the overexpression data should be corroborated by IP studies on native/endogenous interactions in MDP stimulated or infected macrophages.

Answer: We have confirmed the interaction of GEF-H1 and IRF5 in primary macrophages Figure 1d. All functional data was confirmed in primary macrophages isolated from WT, ARHGEF2, IKK ϵ , IRF5, NOD2 and STING deficient mice (Figure 1a, 1b, 1c, 1d 1f and 1g, Figure 2b, 2c, 2e, 2f, and 2g, Figure 4e, Figure 5a, 5b, and 5c, Figure 6b, 6c, and 6d, Supplemental Figure S1) There are limitation for immunoprecipitation studies in primary cell that are associated with level of protein expression, subcellular location, microtubule association and redistribution of activated proteins. However we were able to demonstarte immune complexes between IRF5 and GEFH1 in primary macrophages (Figure 1d).

Critique 1: It is important to more thoroughly examine the role of GEF-H1 in innate immune responses of macrophages/dendritic cells to bacteria such as *Listeria* in addition to the studies in Fig 5. It is recommended that the authors examine the expression of additional immune and inflammatory genes at the RNA and protein levels in WT and GEF-H1-deficient macrophages during *Listeria* infection. Similarly, the authors should perform in vivo studies utilizing GEF-H1-deficient mice to demonstrate if GEF-H1/IKK ϵ /IRF5 axis contribute to host defense responses and bacterial control during infections. This would substantially improve the significance of this study.

Answer: We expanded the experiments that demonstrate that IRF5 activation by *L. monocytogenes* is impaired in the absence of GEF-H1 or IKK ϵ . New experiment show that the regulation of genes that are specifically induced by MDP in a GEF-H1 and IRF5 dependent manner in macrophages, also require GEF-H1, IKK ϵ and IRF5 during *L. monocytogenes* infection (Figure 6d).

Additional comments

Critique 2: Fig 2b and 4f. The levels of total IRF5 itself were lower in KO and inhibitor treated lysates, which could explain the reduction in pIRF5 levels in the corresponding lanes.

In many instances the authors' claim were inconsistent with the data presented.

- Fig 1a. pIRF3 and pIRF5 were present at similar levels in WT cells. But authors claim that MDP activated IRF5 but not IRF3.

Answer: Our data distinguishes between gene expression and functional activation of signaling mediators by phosphorylation. Expression of IRF5 is induced by MDP, but its activation by phosphorylation is impaired in the absence of GEF-H1 (Figure 1a, 1c) with the subsequent disruption of IRF5 dependent gene expression signatures (Figure 5a, 5b and 5c). While IRF5 is responsible for the majority of gene expression induced by MDP, the RNA sequencing results show that additional transcriptional responses occur that are GEF-H1 independent and may regulate the expression signaling intermediaries such as IRF5. As IRF3 is not activated by MDP (supplemental Figure S1) we removed these data set from the figure 1.

We have included new control experiments to clarify the level of TBK1 and IRF3 activation by comparing its phosphorylation by MDP and cyclic di GMP in macrophages in WT and STING deficient macrophages (Supplemental figure S1).

Critique 3: Fig 2g. pIRF5 levels were lower in LPS-treated GEF-H1 KO cells compared to WT cells. But the authors claim that GEF-H1 was specifically required during MDP induced IRF5 activation.

Answer: GEF-H1 is required for IRF5 activation during MDP stimulation. In contrast, IRF5 activation in the TLR4 pathway can occur in the absence of GEF-H1 (Figure 1g). IRF5 is efficiently phosphorylated in GEFH1 deficient macrophages compared to the responses in IKK ϵ deficient macrophages (Figure 1g). Also gene expression in response to TLR4 signaling is not significantly altered in GEF-H1 deficient macrophages as previously shown¹.

Critique 4: Fig 3g. S399A mutant appears to be defective for IRF5 recruitment.

Answer: We determined whether modifying S324 or S399 alone or in combination with S886 of GEF-H1 controlled IRF5 binding. Removing S886 resulted in enhanced IRF5 binding to GEF-H1 (Figure 3f). IRF5 binding was further enhanced to the GEF-H1 variant lacking both S886 and S324 (Figure 3f). In contrast, eliminating S399 in the S886 variant failed to enhance IRF5 binding to GEF-H1 (Figure 3f). Together, these data demonstrated that the innate immune function of GEF-H1 is controlled by dephosphorylation events. Both, S324 and S886 of GEF-H1 are critical in the control of the interaction with IRF5.

Critique 5: It is not clear how the authors have determined that GEF-H1 localized to microtubules. Similarly, the confocal data is not convincing that GEF-H1 was released from microtubules. This should be corroborated using additional approaches as well as in macrophages upon MDP stimulation.

Answer: The association of GEF-H1 has been extensively demonstrated in previous studies^{1, 2}.

Critique: 6 Fig 1f. Does GEF-H1 pull down NOD2 in the absence of IKK ϵ ?

Answer: This is an interesting avenue of investigations we will pursue in future experiments. We were unable to identify antibodies that reliably detect mouse NOD2. 3 different anti-NOD2 antibodies raised against human NOD2 that we tested in WT and NOD2 deficient macrophages either failed to detect or cross-reacted with other proteins of comparable size to NOD2.

Critique 7: Is GEF-H1 constitutively phosphorylated at S399 in macrophages?

Answer: We do not have a specific antibody to confirm this regulatory event in macrophages. Our studies thus far demonstrate that dephosphorylation of S886 is a prerequisite for allowing interaction with IKK ϵ (New Figure 3f). S324 is further critical, as its removal dramatically enhances binding to IKK ϵ and IRF5 (Figure 3f). The S324A/S886A variant of GEF-H1 also remarkably increases phosphorylation of ROCK1 (New Figure 4b). S399 is a critical residue that controls phosphorylation at S886 and thus the inactivation of GEF-H1 (Figure 3b and 3c). The identification of the critical serines in GEF-H1 were possible through experiments with double mutants of GEF-H1 that cannot be inactivated by conformational changes through phosphorylation of S886.

1. Chiang, H.S. *et al.* GEF-H1 controls microtubule-dependent sensing of nucleic acids for antiviral host defenses. *Nat Immunol* **15**, 63-71 (2014).
2. Krendel, M., Zenke, F.T. & Bokoch, G.M. Nucleotide exchange factor GEF-H1 mediates cross-talk between microtubules and the actin cytoskeleton. *Nat Cell Biol* **4**, 294-301 (2002).

Reviewers' comments:

Reviewer #1 (Remarks to the Author):

Corresponding Author: Hans-Christian Reinecker

Title: Microbial Recognition by GEF-H1 Controls IKK ϵ Mediated Activation of IRF5

The authors have resubmitted a previously reviewed manuscript. The essential identification of a GEF-H1 dependent sensor of MDP as a novel component of the innate immune response signaling through IRF5 is exciting and merits publication. The revised manuscript is improved but still suffers from deficiencies in data quality, and quantification of western blots, quantification of immunofluorescence. There remains unanswered questions regarding how the GEF-H1 sensor is triggered by MDP or its receptor and the role of microtubules in controlling this response.

Comments:

Figure 1a pIRF5 blot is cut off. Please show the uncut version of the blot so all immune-reactive bands can be seen.

The authors use the term "IRF5 activation" unless a transcription factor activity assay is demonstrated. Rather, the authors could report on the phosphorylation status of IRF5 without reference to its "activity".

Why are IRF5 protein levels increased with MDP stimulation?

Figure 1d: Figure 1a shows a sharp band of GEF-H1 by western blot, while the immunoblot of IRF5 IPs shows GEF-H1 as a fuzzy band at around 120 kDa. Do the authors have an explanation of this difference? The WCL GEF-H1 blot is generally poor quality and should be improved prior to publication.

Fig 1e: The MW marker on GEF-H1 is shown as 50 kDa rather than 100kDa. This should be corrected

The authors state: "Together these data demonstrated that GEF-H1 acted as an essential NOD2 independent platform assembling RipK2, IKK ϵ , IKK β for the recruitment and activation of IRF5 during MDP recognition". Yet the authors have not shown whether GEF-H1 interacts with these proteins as a series of binary complexes or as a single large complex. Do these proteins co-elute in a glycerol gradient of cytoplasmic extract?

Fig 2e: This figure and all figures which claim to show differences in phosphorylation levels or protein expression should be rigorously quantified.

Fig 3: Phosphorylation at S886 creates a 14-3-3 binding site. Mutation of this site may alleviate competition for this site for IRF5. It is not clear how mutation of S399 controls phosphorylation at other sites on GEF-H1. There could be many different potential mechanisms that explain this observation. Can the authors show that synthetic peptides that contain S886 and or S324 are sufficient to bind to IRF5 or other members of this complex? These mutagenesis studies could be perturbing GEF-H1 structure such that the binding events with IRF5 at distant sites on the protein.

4a: The only magnification bar in these images is provided for the extended focus image. The magnification of the other images should be provided as well as quantification of multiple images showing these effects. I am confused by this figure. Firstly, GEF-H1 is massively overexpressed leading to MT bundles (highly pathological structure) with little colocalization with IRF5. The addition of IKKe leads to redistribution of both GEF-H1 and IRF5. It is difficult to conclude exactly where IRF5 is in these images as it looks like it is aggregated or focused in some sort of subcellular structure or vesicle including diffuse nuclear staining. Does IRF5 Chip to its specific target promoters in the presence of GEF-H1? The authors state "Cells that demonstrated nuclear translocation of IRF5 also exhibited membrane blebbing indicative of Rho GTPase function that is enhanced by GEF-H1 and leads to Rho-associated protein kinase (ROCK1) activation" Membrane blebbing can also be indicative of dying cells. Are these stably transfected cells or transiently transfected? Are these cells alive or dying?

4d. Should show ROCK levels.

4e. shows the phosphorylation of IRF5 at 1 hour in the presence of the ROCK inhibitor, arguing against the role of ROCK in acute phosphorylation of IRF5. Is the effect of ROCK on IRF5 mediated by ROCK1 or ROCK2? Is IRF5 pS445 transcriptional more active than the non-phosphorylated form? An IRF5 promoter reporter assay +/- ROCK1/2 would be very informative.

Figure 5. Triplicate RNAseq measurements are ideally needed for reliable assertions about differences in RNA abundance between two different conditions.

GEF-H1 activation stimulates focal adhesion formation and stress fibers. To what extent is the anti-microbial function of GEF-H1 dependent on intact actin cytoskeleton? Does nocodazole stimulate pIRF5 (as previously requested)?

Does GEF-H1 bind directly to Pglyrps, the receptor for MDP? How does MDP through Pglyrps change the phosphorylation status of GEF-H1 (eg dephosphorylation of S886)? Are specific phosphatases recruited to the complex?

Reviewer #2 (Remarks to the Author):

The authors have address all my comments in the new version of the manuscript.

Line 114:

- IFR5 should be replace by IRF5

-The authors' state: "this indicate that GEF-H1 enabled IKKe to function as an IRF5 kinase".

Did the authors mean "upstream" kinase?

Reviewer #3 (Remarks to the Author):

This reviewer's comments have been sufficiently addressed by the authors through several additional experiments. Overall, the revised version is much improved and reports a novel finding.

Microbial Recognition by GEF-H1 Controls IKK ϵ Mediated Activation of IRF5

Detailed response to the comments:

Reviewers' comments:

Reviewer #1 (Remarks to the Author):

Corresponding Author: Hans-Christian Reinecker

Title: Microbial Recognition by GEF-H1 Controls IKK ϵ Mediated Activation of IRF5

The authors have resubmitted a previously reviewed manuscript. The essential identification of a GEF-H1 dependent sensor of MDP as a novel component of the innate immune response signaling through IRF5 is exciting and merits publication. The revised manuscript is improved but still suffers from deficiencies in data quality, and quantification of western blots, quantification of immunofluorescence. There remains unanswered questions regarding how the GEF-H1 sensor is triggered by MDP or its receptor and the role of microtubules in controlling this response.

Comments:

Question 1: Figure 1a pIRF5 blot is cut off. Please show the uncut version of the blot so all immune-reactive bands can be seen. The authors use the term "IRF5 activation" unless a transcription factor activity assay is demonstrated. Rather, the authors could report on the phosphorylation status of IRF5 without reference to its "activity".

Answer: We have added the complete blot. The phosphorylation of IRF5 is required for the transport of IRF5 to the nucleus to induce transcriptional responses and thus, its activity (Proc Natl Acad Sci U S A. 2014 Dec 9;111(49):17438-43 and Proc Natl Acad Sci U S A. 2014 Dec 9;111(49):17432-7). We have replaced activation with phosphorylation throughout the text.

Question 2: Why are IRF5 protein levels increased with MDP stimulation?

Answer: To define the transcriptional mechanisms that control IRF5 expression during MDP signaling, we would need to initiate a new project focused on analyzing the IRF5 promoter interacting transcription factors in macrophages and delete the candidates in vivo to determine which one is required for induction in this pathway.

Question 3: Figure 1d: Figure 1a shows a sharp band of GEF-H1 by western blot, while the immunoblot of IRF5 IPs shows GEF-H1 as a fuzzy band at around 120 kDa. Do the authors have an explanation of this difference? The WCL GEF-H1 blot is generally poor quality and should be improved prior to publication.

Answer: GEF-H1 is highly phosphorylated in macrophages and there maybe more post translational modification that need to be yet characterized. Buffer conditions and gel systems utilized were different for Western blots and immunoprecipitations and result in different migration of proteins. The IP sample was run longer thus showing the distinct GEF-H1 modifications.

Question 4: Fig 1e: The MW marker on GEF-H1 is shown as 50 kDa rather than 100kDa. This should be corrected

Answer: We corrected the misaligned labeling.

Question 5: The authors state: "Together these data demonstrated that GEF-H1 acted as an essential NOD2 independent platform assembling RipK2, IKK ϵ , IKK β for the recruitment and activation of IRF5 during MDP recognition". Yet the authors have not shown whether GEF-H1 interacts with these proteins as a series of

binary complexes or as a single large complex. Do these proteins co-elute in a glycerol gradient of cytoplasmic extract?

Answer: We used 'platform' to describe the apparent ability of GEF-H1 to assemble multiprotein complexes. We have modified this sentence to read: "Together these data demonstrated that GEF-H1 interacted with RipK2, IKK ϵ , and IKK β for the recruitment and phosphorylation of IRF5 during MDP recognition".

Question 6: Fig 2e: This figure and all figures which claim to show differences in phosphorylation levels or protein expression should be rigorously quantified.

Answer: We agree that quantitation is required for small differences in protein expression or phosphorylation. We have included density gradient analysis in Figure 2e and 2f and included the quantification of nuclear translocation of IRF5 in Figure 4a.

Question 7: Fig 3: Phosphorylation at S886 creates a 14-3-3 binding site. Mutation of this site may alleviate competition for this site for IRF5. It is not clear how mutation of S399 controls phosphorylation at other sites on GEF-H1. There could be many different potential mechanisms that explain this observation. Can the authors show that synthetic peptides that contain S886 and or S324 are sufficient to bind to IRF5 or other members of this complex? These mutagenesis studies could be perturbing GEF-H1 structure such that the binding events with IRF5 at distant sites on the protein.

Answer: Our data thus far demonstrates that enhanced phosphorylation of S886 prevents binding of IKK ϵ . We would submit that phosphorylation of S886 changes the conformation of GEF-H1 to prevent binding to of IKK ϵ . These are certainly interesting studies, which we will attempt in the future. The outcome of the proposed studies is quite unpredictable as we don't know what part of GEF-H1 the synthetic peptides should contain to facilitate or inhibit binding. This would be an extensive undertaken.

Question 8: 4a: The only magnification bar in these images is provided for the extended focus image. The magnification of the other images should be provided as well as quantification of multiple images showing these effects. I am confused by this figure. Firstly, GEF-H1 is massively overexpressed leading to MT bundles (highly pathological structure) with little colocalization with IRF5. The addition of IKK ϵ leads to redistribution of both GEF-H1 and IRF5. It is difficult to conclude exactly where IRF5 is in these images as it looks like it is aggregated or focused in some sort of subcellular structure or vesicle including diffuse nuclear staining. Does IRF5 Chip to its specific target promoters in the presence of GEF-H1? The authors state "Cells that demonstrated nuclear translocation of IRF5 also exhibited membrane blebbing indicative of Rho GTPase function that is enhanced by GEF-H1 and leads to Rho-associated protein kinase (ROCK1) activation" Membrane blebbing can also be indicative of dying cells. Are these stably transfected cells or transiently transfected? Are these cells alive or dying?

Answer: We have added the reference bar to all confocal figures, although the scale of the extended image is the same as that of the single section images that are part of the extended image. We have included a quantification of nuclear IRF5 in Figure 4a. In the presence of kinase deficient IKK ϵ , IRF5 remains in the cytosol and GEF-H1 associated with microtubules. Upon expression of functional IKK ϵ GEF-H1 relocates to the cytoplasm and IRF5 can transition to the Nucleus. We do not know currently the specifics or components of the intracellular compartment that enriches IRF5. This is a very important question that could be addressed in a proteomics approach. It is possible that the continuous activation of ROCK1 in overexpression experiments will be eventually activate pyroptosis or apoptosis in these cells. This is a limitation of experiments using overexpression systems in general. However, the system allows us to determine the activation of ROCK1 in the presence of GEF-H1 variants in Figure 4b and IRF5 phosphorylation in response to ROCK1 and IKK ϵ in Figure 4c. Further, inhibiting ROCK1/2 in primary macrophages inhibits IRF5 phosphorylation in response to MDP (Figure 4e).

Question 9: 4d. Should show ROCK levels.

Answer: We have carried out these localization experiment by live cell imaging as fixation of cells easily disrupts the microtubule network and makes it difficult to accurately follow protein distributions of GEF-H1.

Question 10: 4e. shows the phosphorylation of IRF5 at 1 hour in the presence of the ROCK inhibitor, arguing against the role of ROCK in acute phosphorylation of IRF5. Is the effect of ROCK on IRF5 mediated by ROCK1 or ROCK2?

Answer: The inhibitor we are using can inhibit both ROCK1 and ROCK2 and thus it is possible that ROCK2 could play a role as well. Both kinases are highly similar and no difference in affinity of the Rho binding domain of ROCK1 and ROCK2 for RhoA, RhoB, or RhoC has been described. However cell type specific distinct subcellular distributions and activation mechanisms could result in distinct roles GEF-H1 dependent innate immune activation. We have added this point to the discussion.

Question 12: Is IRF5 pS445 transcriptional more active than the non-phosphorylated form? An IRF5 promoter reporter assay +/- ROCK1/2 would be very informative.

Answer: The phosphorylation of S445 is required for transcriptional activity of IRF5 (Proc Natl Acad Sci U S A. 2014 Dec 9;111(49):17438-43 and Proc Natl Acad Sci U S A. 2014 Dec 9;111(49):17432-7, (called S462 in this paper as they used isoform2 of IRF5)). It is unclear which region of a promoter of a gene that is regulated by IRF5 could be used in these assays. These experiments are interesting to define the precise post-translational modification that convey DNA binding of IRF5 but beyond the scope of the current publication.

Question 13: Figure 5. Triplicate RNAseq measurements are ideally needed for reliable assertions about differences in RNA abundance between two different conditions.

Answer: For in vitro stimulated cells, two samples are sufficient to obtain statistically significant results in the utilized DNA nexus analysis platform that was developed for the Encode consortium (www.dnasnexus.com). Furthermore, the results from the RNA sequencing analysis were independently confirmed by RT-PCR and the analysis of knockout mice.

Question 14: GEF-H1 activation stimulates focal adhesion formation and stress fibers. To what extent is the anti-microbial function of GEF-H1 dependent on intact actin cytoskeleton?

Answer: It is tempting to speculate that GEF-H1 may indeed contribute to the crosstalk between the actin cytoskeleton and the microtubule network. This is another interesting area of investigation we are pursuing as many pathogens target the actin cytoskeleton through specific effectors to facilitate uptake or movement within cells. We expect that possible mechanisms in which GEF-H1 plays a role would be pathogen specific and therefore cannot be generalized.

Question 15: Does nocodazole stimulate pIRF5 (as previously requested)?

Answer: Microtubule stabilizers such as Nocodazole prevent release of GEF-H1 from microtubules and thus prevent GEF-H1 induced immune responses (Nat Immunol. 2014 Jan;15(1):63-71).

Question 16: Does GEF-H1 bind directly to Pglyrps, the receptor for MDP? How does MDP through Pglyrps change the phosphorylation status of GEF-H1 (eg dephosphorylation of S886)? Are specific phosphatases recruited to the complex?

Answer: These are exciting questions. However we failed to identify commercial antibodies against mouse Pglyrp1 that actually work and we are in the process of generating new antibodies. It is not clear whether Pglyrps can bind MDP. Also, we would caution that while this may be an interesting candidate it is rather uncertain if Pglyrp1 as a secreted protein can function as an intracellular receptor for MDP in conjunction with GEF-H1. We are working on this question but a definitive answers may be many months away and would certainly warrant an independent report.

Reviewer #2 (Remarks to the Author):

The authors have address all my comments in the new version of the manuscript.

Line 114:

- IFR5 should be replace by IRF5

-The authors' state: "this indicate that GEF-H1 enabled IKK ϵ to function as an IRF5 kinase".

Did the authors mean "upstream" kinase?

Answer: We have made these corrections.

Reviewer #3 (Remarks to the Author):

This reviewer's comments have been sufficiently addressed by the authors through several additional experiments. Overall, the revised version is much improved and reports a novel finding.